# CBFLOWNET: GENERATING HIGHER-QUALITY CANDIDATES VIA COMBINATORIAL BANDITS

## ABSTRACT

As a probabilistic sampling framework, Generative Flow Networks (GFNs) show strong potential for constructing complex combinatorial objects through the sequential composition of elementary components. However, existing GFlowNets often suffer from excessive exploration over vast state spaces, leading to over-sampling of low-reward regions and convergence to suboptimal distributions. Effectively biasing GFlowNets toward high-reward solutions remains a non-trivial challenge. In this paper, we propose CBFlowNet, which integrates a combinatorial multi-armed bandit (CMAB) framework with GFN policies. The CMAB component prunes low-quality actions, yielding compact subspaces for exploration. Restricting GFNs to these compact subspaces accelerates the discovery of high-value candidates, while the reduced complexity enables faster convergence. Experimental results on multiple tasks demonstrate that CBFlowNet generates higher-reward candidates than existing approaches, without sacrificing diversity. All implementations are publicly available at https://anonymous.4open.science/r/CBFlowNet-E0BA/.

## 1 INTRODUCTION

Generative Flow Networks (GFlowNets) (Bengio et al., 2021; Zhang et al., 2022; Cretu et al., 2024) have shown their impressive potential in generating diverse and high-scoring candidates across various domains, especially in generating combinatorial objects (Zhang et al., 2023c;b). By unifying MDPs' sequential dynamics with flow-based probability matching, GFlowNets synthesize action trajectories that sample candidates proportionally to the desired reward distribution.

Typically, GFlowNets model the generation process as a traversal on a directed acyclic graph (DAG) $(\mathcal{S}, \mathcal{A})$. $\mathcal{S}$ represents the state space (set of nodes) and $\mathcal{A} = \{(s \to s') \mid s, s' \in \mathcal{S}\}$ denotes the set of possible state transitions (set of edges). The key objective of GFlowNets is to sample terminal states (candidates) with probability proportional to a given reward function $R(x)$, i.e., $P(x) \propto R(x)$. This is achieved by learning a flow network, where $F(s)$ represents the total flow through state $s$ and $F(s \to s')$ denotes the edge flow for action $s \to s'$. The forward policy $\pi(s \to s'|s)$, which governs the generation process, is defined as the normalized edge flow: $\pi(s \to s'|s) = \frac{F(s \to s')}{F(s)}$.

However, although the original objective of GFlowNet is to sample candidates in proportion to their rewards, prior studies report that GFlowNet often struggles to generate high-scoring candidates due to excessive exploration in large search spaces (Kim et al., 2023). Moreover, GFlowNet can converge to distributions with average rewards lower than the target, even after extensive training (Shen et al., 2023). Consequently, effectively biasing the sampling process toward high-reward solutions is non-trivial.

A possible way to alleviate such a situation, i.e., avoiding oversampling from low-reward regions, is training the model to sample proportionally to $R(x)^\beta$, where $\beta \gg 1$ represents an inverse temperature parameter (Malkin et al., 2022a; Lau et al., 2024). Optimizing the inverse temperature parameter $\beta$ presents non-trivial challenges, as its selection critically impacts both the exploration-exploitation balance and training stability in GFlowNets.

Aside from tuning the parameter $\beta$ to control the greediness of GFlowNets, improving the sampling process is also promising to increase the greediness of GFlowNets without worrying about stability issues and mode collapse like $\beta$. Lau et al. (2024) combined the flow with Q values to make a greedier

sampling process. Kim et al. (2023) introduced a local search algorithm to make denser samples of high-scoring regions. Our proposed method also focuses on improving the sampling process.

Unlike prior works, we focus on action pruning during the sampling process to avoid oversampling from low-reward regions. Formally, pruning can be defined as selecting $K$ actions to remove from total $N$ actions. Different pruning strategies induce different subspaces, some of which contain denser clusters of high-reward candidates than others. However, pruning is non-trivial: when $K$ scales with $N$, the number of combinations $\binom{N}{K}$ grows exponentially, causing a combinatorial explosion that makes exhaustive search infeasible. With only a limited budget of sampling steps, it becomes essential to efficiently locate promising subspaces, which inevitably introduces an exploration–exploitation trade-off analogous to that in CMAB, where one must decide between exploiting the currently identified high-reward subspaces and exploring alternative ones that may yield even better candidates.

Addressing this challenge, we combine a **combinatorial multi-armed bandit algorithm (CMAB)** framework with GFlowNets, and introduce CBFlowNet. By considering actions as arms in the Multi-armed Bandit problem, we can utilize the CMAB algorithm to select actions that are more likely to lead to high-reward candidates. The combination of strategic pruning with CMAB's exploration mechanism creates a powerful synergy. The pruning of low-quality actions by CMAB results in a greedier sampling strategy that prioritizes high-scoring actions. This bias systematically directs exploration toward higher-quality subspaces, where promising candidates are denser, thus improving overall sample efficiency and generation quality. By focusing on such compact subspaces, CBFlowNet accelerates the learning of high-

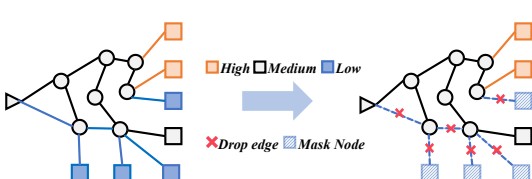

Figure 1: **Illustration of action pruning**. The triangle means initial state, the circles denote interior states, and the squares denote the terminal states. By pruning low-scoring actions(blue edges), candidates with low rewards(blue nodes) are masked. Candidates with high rewards(Orange ones) are more likely to be explored, addressing the over-exploration of low-reward candidates.

reward candidates, as the reduced complexity allows faster convergence. This efficiency extends to other subspaces due to the overlaps of sub-regions. While the pruning focuses the search on promising subspaces, the CMAB component ensures that potentially valuable but under-explored subspaces continue to receive attention.

We evaluate the proposed CBFlowNet on several popular tasks used in prior works, including molecule design (Bengio et al., 2021), three RNA design tasks (Sinai et al., 2020) and bit sequence task (Malkin et al., 2022a). The result demonstrates that the proposed method discovers more high-reward candidates and converges faster than baselines.

## 2 RELATED WORK

**GFlowNets:** Since their introduction by Bengio et al. (2021), GFlowNets have advanced rapidly in theory and applications, with recent works establishing connections to variational inference (Malkin et al., 2022b), distributional analysis (Silva et al., 2025), proxy-free training in offline settings (Chen et al., 2025), and alternative loss designs (Hu et al., 2024). They have also been applied to combinatorial optimization tasks, including general problems (Zhang et al., 2023b), computation graphs (Zhang et al., 2023a), hierarchical exploration with evolutionary search (Kim et al., 2024b), and multi-objective optimization (Zhu et al., 2023). In contrast to these approaches, which mainly employ GFlowNets to solve combinatorial problems, our method leverages combinatorial optimization techniques to improve GFlowNets themselves. Complementary efforts have enhanced GFlowNet training through Q-value integration (Lau et al., 2024), local search (Kim et al., 2023), Thompson sampling (Rector-Brooks et al., 2023), replay strategies (Vemgal et al., 2023; Shen et al., 2023), genetic and evolutionary algorithms (Kim et al., 2024a; Ikram et al., 2024), and adaptive teacher mechanisms (Kim et al., 2024c).

**Combinatorial multi-armed bandit:** The combinatorial multi-armed bandit (CMAB) framework was first introduced by (Chen et al., 2013). Chen et al. (2016) later extended it to nonlinear reward functions dependent on variable distributions. Subsequent work includes the cost-aware auction-based

CMAB by Gao et al. (2021) and full-bandit algorithms by Agarwal et al. (2021); Fourati et al. (2024b), where no individual arm feedback is available.

## 3 PRELIMINARY

In the classic Combinatorial Multi-Armed Bandit (CMAB) problem (Chen et al., 2013; 2016), there are $N$ base arms, each associated with an unknown reward distribution. In every round, the player selects a subset of $K$ base arms, forming a super arm, and receives a joint reward. The goal is to identify the best $K$ arms that maximize the joint reward, which may be a non-linear function of individual arm rewards, while minimizing regret—the gap between the expected reward of always playing the optimal super arm and that of the algorithm's choices (Chen et al., 2013; Slivkins et al., 2019; Chen et al., 2016). The central challenge lies in balancing exploration (trying diverse super arms to gather information) and exploitation (selecting the current best super arm for higher reward). Notably, the flow network embodies a similar dilemma: some regions of the state space contain dense clusters of high-reward candidates, and the algorithm must decide between probing new promising subspaces and exploiting those already identified. The feedback models of CMAB can be categorized into two types: 1) Full-bandit feedback(Chen et al., 2013; Fourati et al., 2024a), where only the aggregate reward of the played super arm is observed, and 2) Semi-bandit feedback(Chen et al., 2013; 2021), where the individual rewards of each base arm in the super arm are additionally revealed. Our problem adopts the semi-bandit setting because the full-bandit feedback would lead to significant information loss regarding the quality of generated candidates. For instance, when a high-scoring candidate is generated by the flow network, it might only utilize a subset of the base arms in the super arm, while the full-bandit setting would obscure this critical information by only providing the aggregated reward, thereby hindering the learning process about which specific base arms contribute most to good solutions.

## 4 METHODOLOGY

In this section, we introduce **Combinatorial Bandit GFlowNet (CBFlowNet)**, a greedy training framework designed to enhance both the quality and diversity of generated candidates based on Combinatorial multi-armed bandit(CMAB). Unlike prior approaches that operate over the entire flow graph (Lau et al., 2024), CBFlowNet achieves a more balanced exploration–exploitation trade-off by selectively focusing on high-scoring subspaces of substantially reduced size.

### 4.1 FRAMEWORK DESIGN

#### 4.1.1 DESIGN OF BASE AND SUPER ARMS

The core challenge in applying CMAB to GFlowNets is to define a set of base arms whose combinations (super arms) can meaningfully constrain the exploration space. A naive approach would be to treat every possible state transition $s \rightarrow s'$ as a distinct arm. However, this leads to an intractably large and state-dependent set of arms, making the CMAB problem ill-posed.

To overcome this, we observe that in many sequential generation tasks, actions can be decomposed into two components: A **state-dependent** component $a_d$ that determines where to act (e.g., which position in a sequence to fill, which molecular stem to extend); A **state-independent** component $a_i$ that determines what action to take, regardless of the specific state (e.g., which value to assign to a position, which building block to attach).

$$\mathcal{A}_i = \{a_i \mid (a_d, a_i) \in \mathcal{A}_s, \forall s \in \mathcal{S}\}. \tag{1}$$

$A_s$ denotes the available transitions(action set) of state $s$. Intuitively, $A_i$ represents the "alphabet" of primitive choices available throughout the generative process. A super arm $\mathbb{S} \subseteq A_i$ is then a subset of this alphabet. By selecting a super arm, we restrict the GFlowNet policy such that at any state $s$, it can only take actions $(a_d, a_i)$ where $a_i \in \mathbb{S}$. This effectively prunes all actions that use primitive choices outside $\mathbb{S}$. Task Examples are given in Table 1.

Table 1: Decomposition of actions into state-dependent and state-independent components across different tasks.

| Task | State-dependent Component ($a_d$) | State-independent Component ($a_i$, Base Arm) | $\mathcal{A}_i$ |
|---|---|---|---|
| Bit Sequence Generation | Position to edit | Binary value to assign | $\{0, 1\}$ |
| Molecule Design | Stem to extend | Building block to attach | Vocabulary of 105 blocks |
| RNA Sequence Design | Prepend or Append | Nucleotide to add | $\{A, C, G, U\}$ |

### 4.1.2 DESIGN OF REWARDS FOR BASE AND SUPER ARMS

We now turn to the assignment of rewards to base arms, a critical factor for ensuring stable learning. In the semi-bandit setting with $m$ base arms, the reward of base arm $i$ at round $t$ is defined as

$$X_i^t = \frac{1}{|C_i^t|} \sum_{x \in C_i^t} r(x), \tag{2}$$

where $C_i^t$ denotes the set of candidates at round $t$ that contain base arm $i$, and

$$r(x) = \text{normalize}(R(x)) \tag{3}$$

is the normalized reward of candidate $x$ with raw reward $R(x)$ from the environment. Normalization ensures all rewards fall within the range $[0, 1]$, which is necessary for effective exploration.

The reward of a super arm is then defined as the sum of the rewards of its constituent base arms.

However, combinatorial bandits (CMAB) require independence among base-arm rewards, which is violated under naive pruning. Formally, if $arm_i = arm'_j$, then $X_i^t \overset{d}{=} X'^t_j$ must hold. Yet, restricting flow networks to a subspace $\mathbb{S}$ artificially inflates the rewards of remaining arms due to the exclusion of low-quality actions.

To address this issue, we adopt a two-phase sampling strategy:

❶ **Constrained training.** Train the flow network restricted to $\mathbb{S}$.

❷ **Unbiased evaluation.** Sample additional candidates without restrictions, and compute $X_i$ as the average reward over all candidates containing arm $i$.

This procedure doubles the sampling cost but does not increase network training complexity, while ensuring unbiased reward estimates. Moreover, the extra cost does not scale linearly with time, since the evaluation stage of deep networks remains unaffected. With multi-threaded sampling, the overhead can be further reduced. A detailed comparison of time consumption is reported in Table 10.

### 4.1.3 DESIGN FOR ADJUSTING NETWORK STRUCTURE

We further propose a method to address the challenge that arises when the independent action space is small but trajectories are long, resulting in narrow yet deep networks.

For instance, in bit sequence generation, there are only two independent actions, $0, 1$, while a complete sequence may require over 100 steps. This leads to a deep but fragile network, where pruning even a single action can collapse the entire solution space.

To address this, we redefine base arms as short sequences of $t$ consecutive actions:

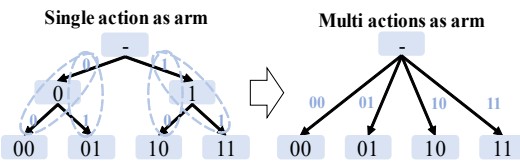

Figure 2: Using short action sequences as arms to transform the narrow-deep network architecture into a more balanced wide structure.

$$a_1 \to a_2 \to \cdots \to a_t.$$

Super arms then consist of sets of such sequences. A sub-trajectory is valid if its independent subsequence belongs to the chosen super arm. This widens the effective search space, balances the architecture, and preserves CMAB guarantees (see Fig. 2).

### 4.1.4 DESIGN OF PROCESS OF THE ALGORITHM

We integrate our framework into the *Combinatorial Upper Confidence Bound (CUCB)* algorithm (Algorithm 1). For each arm $i$, we maintain its empirical mean reward $\hat{\mu}_i$ and selection count $T_i$. The UCB-adjusted estimate is

$$\overline{\mu}_i = \hat{\mu}_i + \sqrt{\frac{3\ln t}{2T_i}}, \tag{4}$$

where $t$ is the round index. This estimate ensures a principled balance between exploration and exploitation.

Training begins with an initialization phase to provide each base arm with a reasonably accurate initial estimate. In subsequent rounds, Figure 3 illustrates the pipeline. The framework first ingests the full DAG, then performs an auxiliary sampling pass to refine reward estimates within the CMAB module. Using these updated estimates, CUCB selects the top-$K$ arms according to $\overline{\mu}_i$, constructs the corresponding super arm. Training restricts sampling to this subspace, and the resulting candidates are used to update the flow network via objectives such as Flow Matching (FM). Within CMAB, the Combinatorial Upper Confidence Bound (CUCB)(as line 11 of Algorithm 1) rule balances exploitation of the current best subspace with exploration of promising alternatives, improving efficiency without compromising diversity.

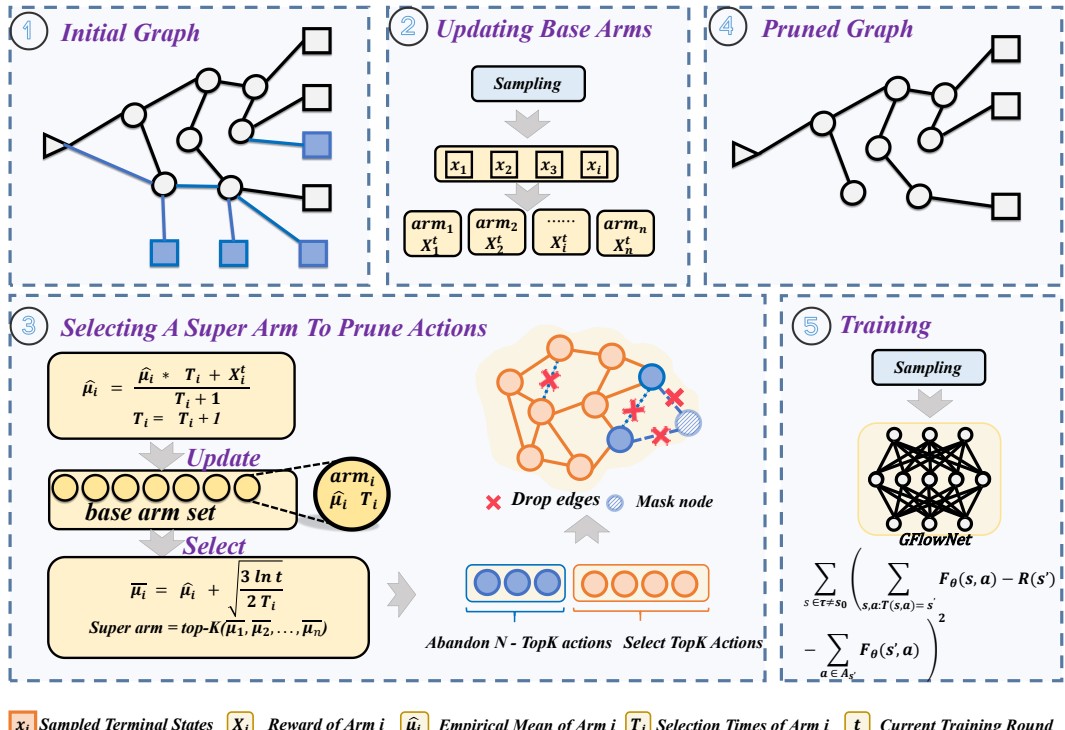

Figure 3: **Illustration of workflow.** The triangle means initial state, the circles denote interior states, and the squares denote the terminal states. Only one round is shown as an example for clarity. The training objective shown is **Flow Matching(FM)** objective.

## 4.2 THEORETICAL FOUNDATIONS OF THE PROPOSED METHOD

### 4.2.1 KEY ASSUMPTIONS OF CMAB IN FLOW NETWORKS

There are two assumptions required by CMAB methods. We show that flow networks naturally satisfy these conditions.

**Monotonicity.** For a super arm $\mathbb{S}$, the expected reward $r_\mu(\mathbb{S})$ is non-decreasing in the reward vector $\mu$. This assumption is natural in flow networks because if all base arms (actions) within a super arm $\mathbb{S}$ exhibit higher expected rewards (i.e., $\mu_i' \geq \mu_i$ for all $i \in \mathbb{S}$), it implies that the flow network under $\mu'$ is better optimized than under $\mu$.

**Bounded Smoothness.** There exists an increasing function $f$ such that

$$|r_\mu(\mathbb{S}) - r_{\mu'}(\mathbb{S})| < f\left(\max_{i \in \mathbb{S}} |\mu_i - \mu_i'|\right). \tag{5}$$

In our framework, $r_\mu(\mathbb{S})$ is the cumulative reward of its arms, reducing $f$ to $f(x) = Kx$ where $K = |\mathbb{S}|$. This ensures stability: small perturbations in individual arm rewards do not cause disproportionate fluctuations at the super-arm level.

### 4.2.2 DYNAMIC NATURE OF FLOW NETWORKS

Unlike standard CMAB problems with stationary distributions, flow networks evolve during training. It is therefore important to characterize this non-stationarity. The fundamental constraint for an ideal flow network is:

$$\pi(x) = \frac{R(x)}{\sum_{x' \in \mathcal{X}} R(x')}, \quad \forall x \in \mathcal{X}, \tag{6}$$

where $\pi(x)$ represents the target distribution and $R(x)$ denotes the reward function. However, achieving this equilibrium condition requires exhaustive exploration of all states, which is impractical in real-world scenarios. Consequently, the expected rewards of individual base arms evolve dynamically throughout the training process of the flow network. Nevertheless, we establish theoretically that these reward distributions converge as training progresses.

**Theorem 1.** For a dynamic flow network where candidates are sampled proportionally to their rewards, the reward distribution of each base arm $i$ converges to a stable distribution. The proof is given in Appendix C.

## 5 EXPERIMENT

We experimented on 5 commonly used standard tasks. As baselines, we use Trajectory Balance(TB)(Malkin et al., 2022a), Sub-Trajectory Balance(SUBTB)(Madan et al., 2023), Detailed Balance(DB)(Jain et al., 2022; Malkin et al., 2022a), LSGFN(Kim et al., 2023), Teacher(Kim et al., 2024c) and QGFN(Lau et al., 2024). We additionally introduced a random algorithm that randomly selects super arms called RandGFN. All experiments are conducted on NVIDIA Tesla A100 80GB GPUs.

### 5.1 BIT SEQUENCE GENERATION

#### 5.1.1 TASK DEFINITION

The task is to generate binary bit sequences using the set $\{0, 1\}$ with a fixed length $n = 120$ with a terminal state space of $2^{120} \approx 10^{36}$ and more intermediate states. The reward of a terminal $x$ is defined as $R(x) = \exp(-\min_{m \in M} \text{dist}(x, m))$, where $\text{dist}(x, m)$ is the Levenshtein Distance of two sequences following (Malkin et al., 2022a; Zhang et al., 2023c). $M$ is a predefined sequence set to be discovered as modes. The mode $m \in M$ is regarded as found if there exists a sample $x$ satisfying $\text{dist}(x, m) < \delta$, where $\delta$ is a predefined parameter.

In this task, we consider a more complex version with many more intermediate states. Malkin et al. (2022a) considers the process as a left-to-right generation where the state space is only a simple tree. Lau et al. (2024); Shen et al. (2023) use a prepend-append MDP to induce a DAG. In our setting, instead of prepending or appending to the existing sequence, we first divide the sequence into $\lfloor \frac{n}{k} \rfloor$ positions. We can insert a generated $k$-bit into any unfilled position, resulting in a more complex DAG.

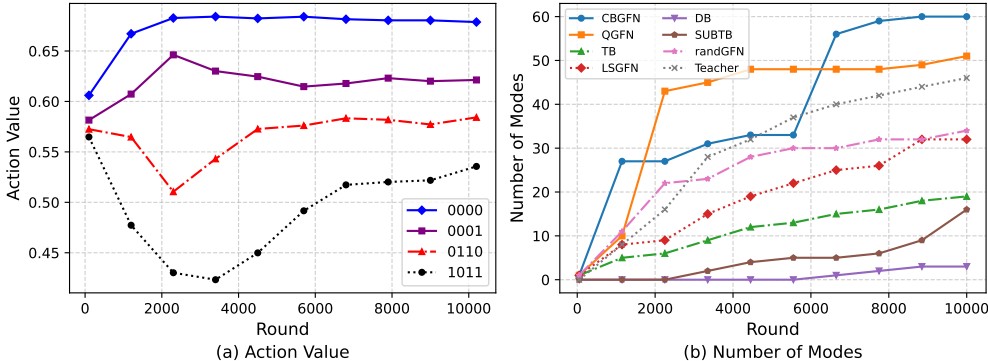

Figure 4: **Experimental results on Bit Sequence task.** Left panel shows how different action values change as the training progresses. Right panel shows the mode discovered by different methods.

### 5.1.2 RESULT

The performance comparison of different methods on the bit sequence task is illustrated in Figure 4 and Table 2. GFlowNets employing TB objective demonstrate superior results, outperforming all other objective functions. CBFlowNet successfully identifies all potential modes, representing a substantial advancement in mode discovery efficiency. Figure 4-a illustrates the evolution of $\mu_i$ for various base arms (actions). As training progresses with the CUCB algorithm's action selection, the arms diverge, converging to distinct outcomes. Notably, the actions $\{0000, 1111\}$ emerge as the highest-scoring, aligning with the construction of $M$, where $\{0000, 1111\}$ appear most frequently compared to other actions.

Table 2: Comparison on Bit Sequence Generation. *Modes* means the number of discovered modes. *Top1000* denotes the average reward of the best 1000 candidates.

| Model | Modes | Top1000 |
|---|---|---|
| **CBGFN** | **60** | **3.60** |
| QGFN | 51 | 3.42 |
| LSGFN | 32 | 2.98 |
| TB | 19 | 2.89 |
| SUBTB | 16 | 2.84 |
| DB | 3 | 2.66 |
| Teacher | 46 | 3.17 |
| RandGFN | 32 | 3.06 |

Table 3: Comparison on Molecule Design. *Modes R>7.5/8* means the number of modes with a reward bigger than 7.5/8. *Top1000* denotes the average reward of the best 1000 candidates.

| Model | Modes R>7.5 | Modes R>8 | Top1000 |
|---|---|---|---|
| **CBGFN** | **13074** | **3207** | **8.436** |
| QGFN | 8567 | 1420 | 8.364 |
| LSGFN | 4514 | 555 | 8.316 |
| TB | 2507 | 308 | 8.233 |
| SUBTB | 6 | 0 | 7.245 |
| DB | 6 | 0 | 7.124 |
| Teacher | 7811 | 1308 | 8.395 |
| RandGFN | 2188 | 282 | 8.248 |

## 5.2 MOLECULE DESIGN

### 5.2.1 TASK DEFINITION

We consider the most common scenario for GFlowNets, the fragment-based molecule generation task. The objective is to design a variety of molecules with a high reward, where the reward is given by a proxy model predicting the binding affinity to the sEH (soluble epoxide hydrolase) protein based on a docking prediction(Trott & Olson, 2010). We use the proxy model provided by (Bengio et al., 2021).

In this task, the states are represented as molecule graphs or SMILES [1]. The action space consists of two components: selecting which molecular stems to extend and choosing which building blocks to add. The maximum number of allowed blocks controls the size of the state space. The vocabulary of building blocks consists of 105 distinct elements, where a block has several possible attachment points(stems). We generate a molecule graph of up to 8 fragments. Therefore, the terminal state space is more than $105^8 \approx 10^{16}$.

---

[1] https://en.wikipedia.org/wiki/Simplified_Molecular_Input_Line_Entry_System

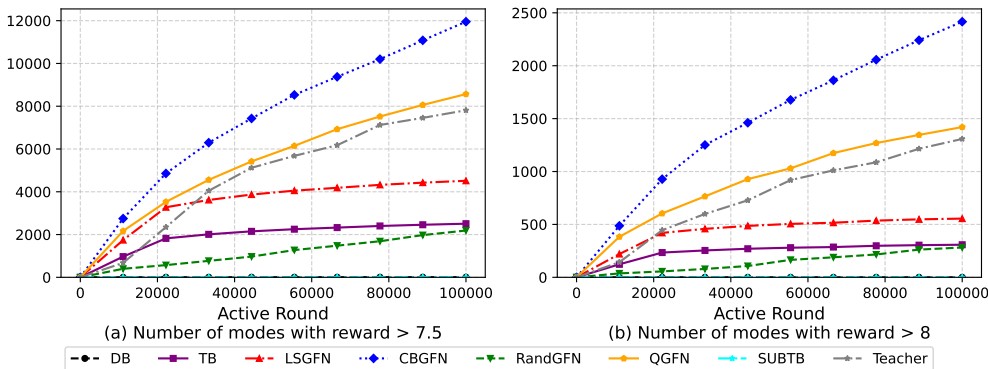

Figure 5: **The curve of number of modes varying with rounds** $(10^3)$ **on Molecule Design.** Left panel shows the number of modes discovered with a reward $R > 7.5$. Right panel shows the number of modes discovered with a reward $R > 8$.

We define each block as a base arm and choose $K$ blocks as a super arm. There are $C_{105}^K$ different base arms. We use Tanimoto similarity (Bender & Glen, 2004) to distinguish different modes, with a threshold of $0.7$. Furthermore, we conduct a comprehensive analysis to examine how different values of K affect the algorithm's performance, particularly in terms of controlling its greediness. The detailed results of this analysis are presented in Appendix F.1. We also include top-K tanimoto similarity as a metric of diversity in Appendix J.

### 5.2.2 RESULT

The comparative performance of various models is summarized in Table 3 and Figure 5. What we found extremely strange is that the baseline SUBTB behaves poorly in both discovering high-scoring modes and generating high-scoring top 1000 candidates. This phenomenon also appears in the molecule design task of Lau et al. (2024). Our proposed CBFlowNet demonstrates remarkable improvements in high-scoring mode discovery. During our experiments, the model successfully identified over 10,000 high-scoring modes (R>7.5) with just 400,000 sampled trajectories (equivalent to 100,000 training rounds). Furthermore, CBFlowNet achieves superior performance in terms of average reward for the top 1000 candidates, outperforming all baseline methods.

### 5.3 RNA-BINDING

#### 5.3.1 TASK DEFINITION

The task is to generate a string of 14 nucleobases. We use a prepend-append MDP to keep adding tokens to a string until it reaches the maximum length, following Kim et al. (2023). There are 4 tokens: adenine (A), cytosine (C), guanine (G), and uracil (U). We conducted experiments on three different target transcriptions: L14-RNA1, L14-RNA2, and L14-RNA3 proposed by Sinai et al. (2020). We treat each token as a base arm, and $K$ is set to $2/3$, denoting that we can either choose 2 base arms or 3 base arms as a super arm.

#### 5.3.2 RESULT

Figure 6 reports the results on three RNA tasks, each evaluated by Number of Modes Discovered, Average Reward, and Top-1000 Reward. CBFlowNet consistently outperforms baselines: it discovers nearly twice as many modes as the strongest competitor and achieves higher average and Top-1000 rewards in Tasks 1 and 2, while remaining competitive in Task 3. The slight instability in Task 3 stems from averaging over only 10 rounds, during which the agent may explore subspaces with suboptimal rewards or insufficiently learned dynamics.

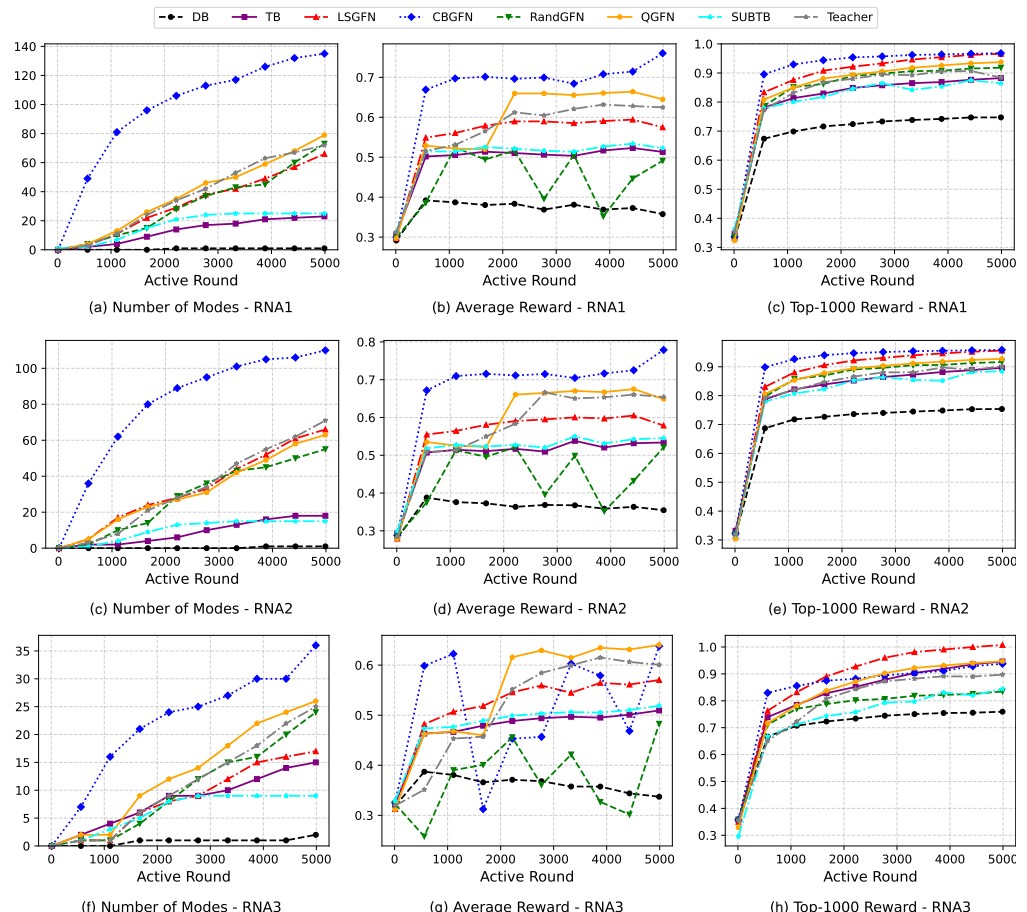

Figure 6: **Performance comparison on RNA design tasks.** Rows correspond to RNA-1, RNA-2, and RNA-3, respectively.

## 5.4 MORE DETAILS

We also reported evidence lower bound (ELBO) as an evaluation of goodness of fit to the target distribution in Appendix L. Besides, we reported some alternative pruning strategies aside from top K actions in Appendix H.

## 6 CONCLUSION AND LIMITATION

**Conclusion:** This paper proposes CBFlowNet, a method designed to enhance the greediness of the sampling process while preserving the diversity of generated candidates. We begin by partitioning the entire state space into multiple subspaces. Next, we employ the CUCB algorithm to effectively balance exploration and exploitation and find the optimal subspaces. To address the challenges posed by narrow-deep network architectures, we propose techniques to transform them into more balanced wide-deep structures. Experimental results across various tasks demonstrate the effectiveness and efficiency of CBFlowNet.

**Limitations:** Although our framework is theoretically applicable to listwise recommendation and combinatorial optimization problems, empirical validation on these tasks remains for future work. Another limitation of the proposed method is that it assumes a fixed reward distribution in the environment. In scenarios where the high-reward state space shifts during training, the benefits are limited and may even disappear.

ETHICS STATEMENT

This research is entirely based on publicly available benchmark environments, including the Bit Sequence task (Malkin et al., 2022a), molecule design task (Bengio et al., 2021), and RNA sequence design tasks (Sinai et al., 2020). These datasets do not contain any personally identifiable or sensitive information, and no human or animal subjects were involved, so no ethical approval was required. While generative modeling frameworks such as GFlowNets and CBFlowNet may have downstream applications in high-stakes domains (e.g., drug discovery or personalized recommendation), the contributions of this work are purely methodological and restricted to controlled benchmark environments. We encourage future applications to carefully assess potential societal impacts, incorporate domain-specific safeguards, and ensure responsible deployment. We declare no conflicts of interest.

REPRODUCIBILITY STATEMENT

We have made all implementations and experimental details publicly available at `https://anonymous.4open.science/r/CBFlowNet-E0BA/`. The repository includes training scripts, model definitions, and configuration files to reproduce the reported results. Hyperparameter choices for all tasks are explicitly documented in the appendix (Tables 5, 6, and 7), along with details of the baselines, evaluation metrics, and hardware setup (NVIDIA Tesla A100 80GB GPUs). We additionally report sensitivity analyses on key parameters (e.g., $K$ for super arms, $\beta$ for reward exponent, $\epsilon$ for exploration) in Appendices F and G. These results ensure that our findings are robust and reproducible across different settings.

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

CONTENTS

## A  LLM USAGE STATEMENT

We only use LLMs as a language optimization tool to polish sentences, improving their readability and fluency. The LLM did not contribute to the scientific ideas, algorithm design, or experimental setup. All substantive content, reasoning, and conclusions are entirely the product of the authors. We accept full responsibility for all content in the paper, including parts refined or corrected by the LLM, and affirm that no text generated by the LLM constitutes original scientific contributions attributed to it.

## B  SYMBOL TABLE

This section provides a summary of the key notations and symbols used throughout this paper. The symbols are listed in Table 4.

Table 4: Notation

| Symbol | Description |
|--------|-------------|
| $\mathcal{S}$ | State space |
| $\mathcal{A}$ | Action space |
| $s_0, s', s$ | Possible states |
| $x$ | Terminal state |
| $\mathbb{S}$ | Super arm |
| $\mathcal{X}$ | Set of terminal states |
| $\beta$ | Inverse temperature parameter |
| $A_i$ | Independent action set |
| $A_s$ | Valid action set of state $s$ |
| $a_i$ | State-independent action component |
| $a_d$ | State-dependent action component |
| $X_i^t$ | the reward of base arm $i$ at round $t$ |
| $C_i^t$ | the set of candidates at round $t$ that contain base arm $i$ |
| $R(x)$ | Reward of terminal state |
| $r(x)$ | Normalized reward of terminal state |
| $\hat{\mu}_i$ | Empirical mean reward of base arm $i$ |
| $T_i$ | Number of times base arm $i$ been selected |
| $\overline{\mu_i}$ | UCB-adjusted reward of base arm $i$ |
| $\mu, \mu'$ | reward vector of base arms |
| $K$ | Number of base arms in a super arm |
| $N$ | Size of base arm set |
| $s \rightarrow s'$ | state transition |
| $F(s)$ | State Flow |
| $F(s \rightarrow s')$ | Edge Flow from state $s$ to $s'$ |
| $Z$ | Flow of the initial state $s_0$ |
| $P_F(s'\|s)$ | Forward transition probability from $s$ to $s'$ |
| $P_B(s\|s')$ | Backward transition probability from $s'$ to $s$ |
| $\mathcal{L}_{\text{FM}}$ | Flow Matching Objective |
| $\pi(s \rightarrow s'\|s)$ | Policy for selecting action $s \rightarrow s'$ at node $s$ |

## C  THEOREM PROOF

**Theorem 1.** For a dynamic flow network where candidates are sampled proportionally to their rewards, the reward distribution of each base arm $i$ converges to a stable distribution.

**Proof .** We establish the convergence in two parts:

**Part 1: Stability under Fixed Flow Network**
When the flow network stabilizes, the reward distribution of arm $i$ becomes stationary. Let:

$$X_{i,t} = \frac{1}{|C_{i,t}|} \sum_{x \in C_{i,t}} R(x), \tag{7}$$

where:

- $C_{i,t} = \{x_1, \ldots, x_{|C_{i,t}|}\}$ denotes candidates sampled at round $t$ containing base arm $i$.
- $\mathcal{X}_i \subset \mathcal{X}$ represents terminal states containing base arm $i$.
- $R(x) : \mathcal{X}_i \rightarrow \mathbb{R}$ follows a fixed distribution.
- Candidates $x \in C_{i,t}$ are generated with probability $p(x) = F(x)/\sum_{x' \in \mathcal{X}_i} F(x')$.

Since $R(x)$ are i.i.d. and $p(x)$ becomes stationary, $X_{i,t}$ converges to a fixed distribution by the Law of Large Numbers.

**Part 2: Convergence under Network Evolution**
As the flow network converges, we have:

$$C_{i,t} \xrightarrow{d} \mathcal{C}_i, \tag{8}$$

where $\mathcal{C}_i$ denotes the limiting candidate distribution. Because:

- $R(x)$ are i.i.d. and depend only on $x$

- The mapping $(C_{i,t}, \{R(x)\}_{x \in C_{i,t}}) \mapsto X_{i,t}$ is continuous in the discrete topology (where any function on a discrete space is continuous).

By the Continuous Mapping Theorem, we obtain:

$$X_{i,t} \xrightarrow{d} X_i = \frac{1}{|\mathcal{C}_i|} \sum_{x \in \mathcal{C}_i} R(x), \tag{9}$$

establishing the desired convergence result.

## D  REGRET ANALYSIS

### D.1  ABSENCE OF OPTIMAL SUPER ARMS

The CMAB framework typically relies on an oracle capable of providing an $(\alpha, \beta)$-approximation of the optimal super arm (i.e., a subset of base arms that maximizes the expected reward) to make a tight analysis of the regret bound. However, in flow networks, identifying the exact optimal super arm is computationally prohibitive due to the combinatorial explosion of possible states. Even an $(\alpha, \beta)$-approximation of the optimal super arm is impractical because the $\alpha, \beta$ might change when the flow network is different.

### D.2  REGRET BOUND

The flow network evolves dynamically throughout training, resulting in time-varying reward distributions for each base arm $i$. Although these distributions are guaranteed to converge asymptotically, the inherent non-stationarity introduces considerable uncertainty into the system. Such temporal variability, together with the absence of clearly defined optimal super arms, presents significant challenges for deriving a precise regret bound for the learning algorithm.

However, we can still construct empirical regret metrics to evaluate algorithm performance. A practical approach is to use the best empirical arm observed up to time $t$, denoted as $\hat{\mu}_t(\mathbb{S}^*)$, as a proxy for the unknown optimal mean reward. The estimated cumulative regret is then computed as:

$$\hat{R}(T) = \sum_{t=1}^{T} (\hat{\mu}_t(\mathbb{S}^*) - \hat{\mu}_t(\mathbb{S})). \tag{10}$$

We employ the metric of empirical cumulative regret to quantify the performance gap between consistently selecting the currently known optimal super arm and the super arm chosen by our algorithm. To comprehensively evaluate our method's efficacy in minimizing regret, we introduce a baseline random strategy called RandGFN that uniformly selects $K$ base arms at each round.

Figure 7 presents the cumulative regret curves for CBFlowNet and the RandGFN, revealing distinct performance differences across tasks:

1. Bit Sequence Generation Task (left panel): The random policy exhibits competitive performance, resulting in a moderate regret gap between the two models. This suggests that simple heuristics suffice in this simpler task setting.

2. Molecule Design Task (middle panel): CBFlowNet demonstrates substantial improvement, achieving a significantly lower cumulative regret than RandGFN. This highlights the proposed model's effectiveness in optimizing structured, complex objectives.

3. L14-RNA1 Task (right panel): The regret gap widens again, where CBFlowNet substantially outperforms RandGFN, indicating its ability to handle intricate combinatorial challenges in nucleic acid sequence optimization.

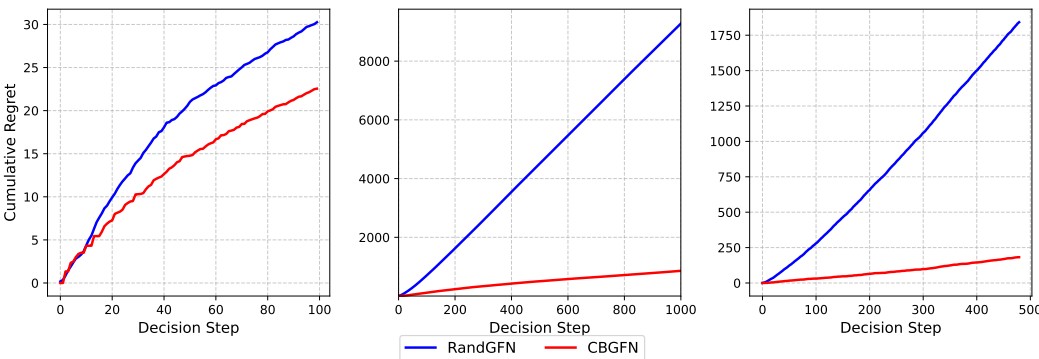

Figure 7: **Experimental result on cumulative regret with different tasks.** Left: Cumulative regret of models in Bit Sequence Generation Task. Center: Cumulative regret of models in Molecule Design Task. Right: Cumulative regret of models in L14-RNA1 Task.

Table 5: Key hyperparameter setting in Bit Sequence Generation task

| Parameter | Value |
|---|---|
| Batch size | 16 |
| Number of steps | 10000 |
| k-bits | 4 |
| Lamda | 1.9 |
| Learning rate | 1e-3 |
| Z Learning rate | 1e-3 |
| $\beta$ | 2 |
| Explore Epsilon | 0.01 |
| K | 4 |
| Decision Interval | 100 |

## E  EXPERIMENT DETAILS: BIT SEQUENCE

Here we present the hyperparameter configuration for our bit sequence generation experiments (Table 5). While adopting the baseline framework from Malkin et al. (2022a), we reduce the training steps from 50,000 to 10,000. Each action is represented by $k = 4$ bits, and through empirical validation, we selected K=4 candidate arms from $\{2, 4, 6, 8, 10\}$. For the CMAB algorithm, we determined 100 steps to be the optimal decision interval after evaluating candidates from $\{50, 100, 200, 300, 400, 500\}$.

### E.1  OPTIMAL SUPER ARM CONFIGURATION

The set $M$ is constructed through random combinations of substrings derived from the base patterns $\{00000000, 11111111, 11110000, 00001111, 00111100\}$. For the case where $K = 4$, we can analytically determine the optimal super arm configuration as $\mathbb{S} = \{0000, 1111, 1100, 0011\}$. This configuration enables perfect mode identification within its substate space, achieving an average distance of $0.0$ to all target modes. We evaluate the performance when consistently selecting this optimal super arm configuration in Figure 8(a).

### E.2  EXPERIMENTS ON DIFFERENT $\epsilon$

We conduct a systematic investigation of the exploration parameter $\epsilon$, which controls the probability of random actions in the GFlowNets framework. This parameter critically influences the sampling behavior of the flow network within substate spaces. As shown in Figure 8(b), through comprehensive testing across $\epsilon$ values $\{0.1, 0.01, 0.02, 0.001, 0.0005\}$, we find that $\epsilon = 0.01$ demonstrates superior performance in terms of mode discovery and sample quality.

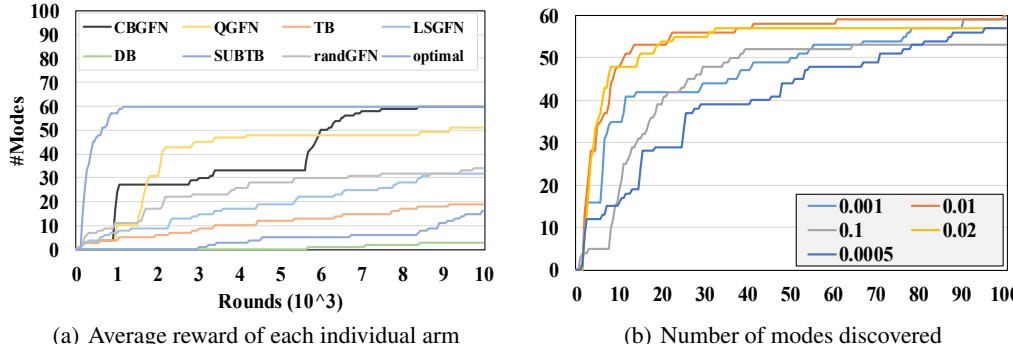

(a) Average reward of each individual arm                (b) Number of modes discovered

Figure 8: **Supplementary Experimental Results for the Bit Sequence Task.** Left: Number of modes identified by CBFlowNet when consistently selecting the optimal super arm. Right: Number of modes identified across varying $\epsilon$ values.

Table 6: Key hyperparameter setting in Molecule Design

| Parameter | Value |
|---|---|
| Batch size | 4 |
| Number of steps | 100000 |
| Lamda | 0.99 |
| Learning rate | 5e-4 |
| $Z$ Learning rate | 5e-3 |
| Tanimoto Similarity Threshold | 0.7 |
| $\beta$ | 8 |
| Explore Epsilon | 0.05 |
| $K$ | 30 |
| Decision Interval | 400 |

The results indicate that moderate exploration ($\epsilon = 0.01$) achieves the best balance between exploration and exploitation, while higher values lead to excessive randomness and lower values result in insufficient exploration of the state space.

## F  EXPERIMENT DETAILS: MOLECULE DESIGN

We present the hyperparameter configurations for our Molecule Design Task experiments, as detailed in Table 6. Building upon the framework established by Bengio et al. (2021), we maintain their default parameter settings while introducing specific optimizations. After empirical evaluation, we set the number of base arms $K$ to 30, selected from $\{8, 10, 20, 30, 40, 50, 60, 70, 80, 90, 100\}$, and determined the optimal decision interval for the CMAB algorithm to be 400 from $\{50, 100, 200, 300, 400, 500\}$.

### F.1  EXPERIMENTS ON DIFFERENT $K$

The hyperparameter $K$, which determines the size of the super arm, serves as our primary mechanism for controlling the method's greediness. In our molecule generation experiments, we evaluate 10 distinct $K$ values ranging from 8 to 100. Smaller $K$ values correspond to greedier selections, as we restrict our choice to only the top $K$ base arms, resulting in more compact sub-state spaces. However, this increased greediness comes at the cost of reduced candidate diversity, as demonstrated in Figure 9. While $K \in \{8, 10, 20\}$ yields higher average scores among the top 1000 candidates, it significantly compromises molecular diversity (measured by the number of distinct modes). Through a comprehensive evaluation, we identify $K = 30$ as the optimal setting for our molecular design task, achieving an effective balance between candidate quality and diversity.

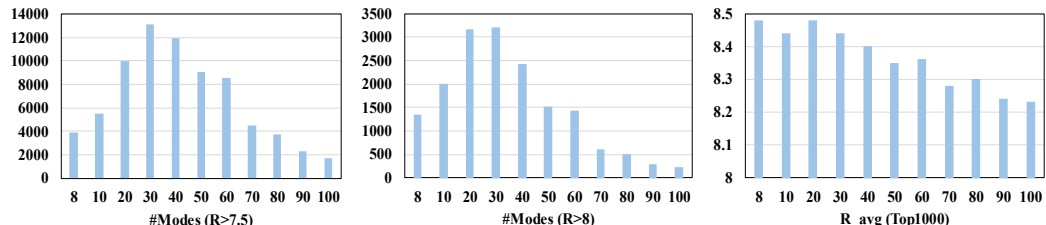

Figure 9: **Experimental result on Molecule Design with different $k$ with 400,000 samples.** Left: The number of modes R>7.5 with a Tanimoto similarity threshold of 0.7. Center: The number of modes R>8 with a Tanimoto similarity threshold of 0.7. Right: The average reward of the top 1000 high-scoring samples.

Table 7: Key hyperparameter setting in RNA-Binding task

| Parameter | Value |
| --- | --- |
| Batch size | 32 |
| Number of steps | 5000 |
| RNA length | 14 |
| MDP style | Prepend and Append |
| Lamda | 0.9 |
| Learning rate | 1e-4 |
| Z Learning rate | 1e-2 |
| Mode metric | Hamming Ball 1 |
| $\beta$ | 20 |
| Explore Epsilon | 0.01 |
| K | 2/3 |
| Decision Interval | 50 |

## G   EXPERIMENT DETAILS: RNA-BINDING

In this section, we give the hyperparameters used for each of our experiments' RNA-Binding Task as shown in Table 7. In our experimental setup, the learning rate of $1 \times 10^{-4}$ is selected from $\{1 \times 10^{-5}, 1 \times 10^{-4}, 1 \times 10^{-3}, 5 \times 10^{-3}\}$ and the Z learning rate of $1 \times 10^{-2}$ is selected from $\{1 \times 10^{-5}, 1 \times 10^{-4}, 1 \times 10^{-3}, 5 \times 10^{-3}\}$. The Lambda for SUBTB uses 0.9 out of $\{0.8, 0.9, 0.99, 0.999\}$. The explore epsilon is used to control the random action probability, five values are tested, including $\{0.1, 0.01, 0.001, 0.0001, 0.0005\}$. We set the reward exponent $\beta$ to 8 from $\{3, 4, 5, 6, 7, 8, 9, 10\}$. $K$ is the number of base arms to compose the super arm selected from $\{1, 1/2, 2, 2/3, 3, 3/4, 4\}$, where $x/x + 1$ denotes we can choose $x$ or $x + 1$ base arms as a super arm. Please note that the RNA1 environment we use is a little different from Teacher(Kim et al., 2024c). Our environment has 1590 modes in total, but Kim et al. (2024c) has 8967 modes. We replace the environment file of Teacher with our environment file to make a fair comparison. The environment is constructed following Kim et al. (2023).

### G.1   DIFFERENT SETTINGS: EXPONENT $\beta$

Since introduced by Bengio et al. (2021), there is already a useful technique to increase the greediness of GFlowNets, that is the exponent $\beta$. The adjusted reward function is formulated as $\hat{R}(x) = R(x)^{\beta}$. The higher $\beta$ makes the model greedier but at the cost of greater numerical instability. Besides, since the middle-reward regions are adjusted into low-reward regions, the diversity is also reduced, leading to mode collapse (Lau et al., 2024). The choice of exponent $\beta$ critically influences the behavior of the CUCB algorithm, as it directly modulates the reward scaling of individual arms. We experimented on different settings of exponent $\beta$ as shown in Figure10, Figure11 and Figure12.

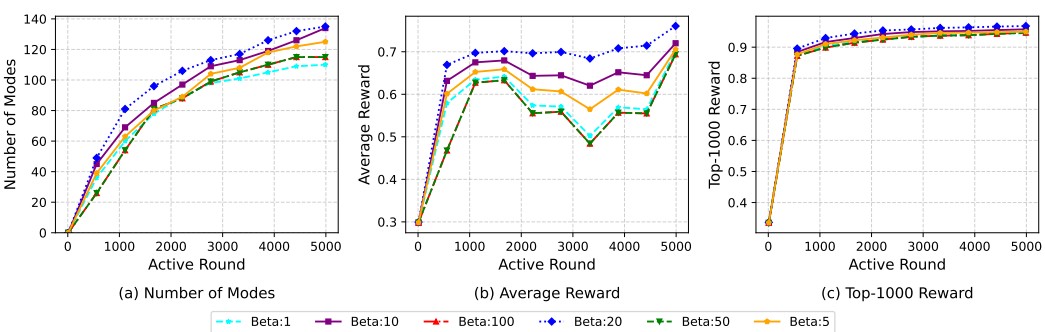

Figure 10: **Experimental result on RNA-Binding Task 1 with different $\beta$.**

In the L14-RNA1 task, models with varying $\beta$ values consistently identified over 100 distinct modes, with the $\beta = 20$ configuration demonstrating superior performance compared to other settings. Notably, the model with $\beta = 1$ exhibited significantly poorer performance relative to other $\beta$ values. Regarding average reward metrics, the $\beta = 20$ model achieved substantially better results, establishing a clear performance gap over other configurations. However, all models showed comparable performance when evaluating the top 1000 rewards.

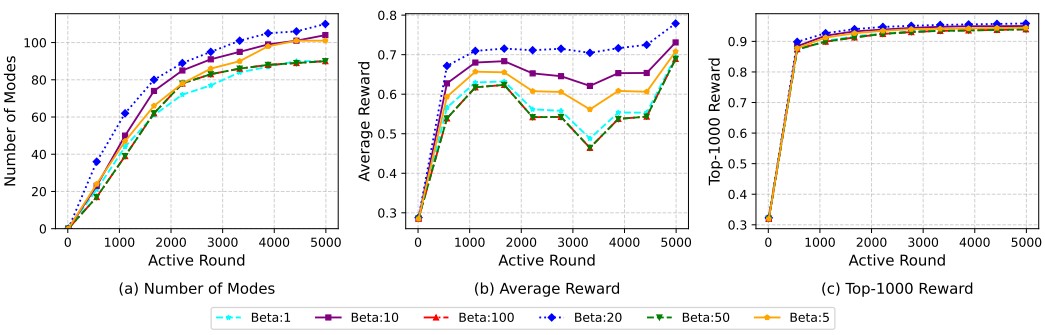

Figure 11: **Experimental result on RNA-Binding Task 2 with different $\beta$.**

In the L14-RNA2 task, while maintaining performance trends consistent with L14-RNA1, the task proved more challenging for mode discovery. All models identified fewer modes compared to L14-RNA1, yet the $\beta = 20$ configuration consistently demonstrated superior performance across all metrics, maintaining its lead over other parameter settings.

L14-RNA3 proves to be the most challenging task, exhibiting a significant decline in both discovered modes and average reward compared to other tasks. In this setting, $\beta = 10$ achieves the best overall performance, whereas $\beta = 20$, while attaining a higher average reward, suffers from instability. The task's difficulty suggests that using an excessively large $\beta$ may be suboptimal, as it risks mode collapse by overly prioritizing high-reward candidates while neglecting mid-reward solutions.

## H   ALTERNATIVE STRATEGIES OF SUPER ARMS

Aside from top-K actions, we came up with three alternatives of arm selection: 1) selecting base arms proportional to their scores; 2) selecting base arms randomly; 3) keeping hard pruning but removing CMAB. Here, hard pruning refers to the permanent removal of some fixed actions from the selection pool. Random selection assigns equal selection probabilities to all base arms when constructing the super arm.

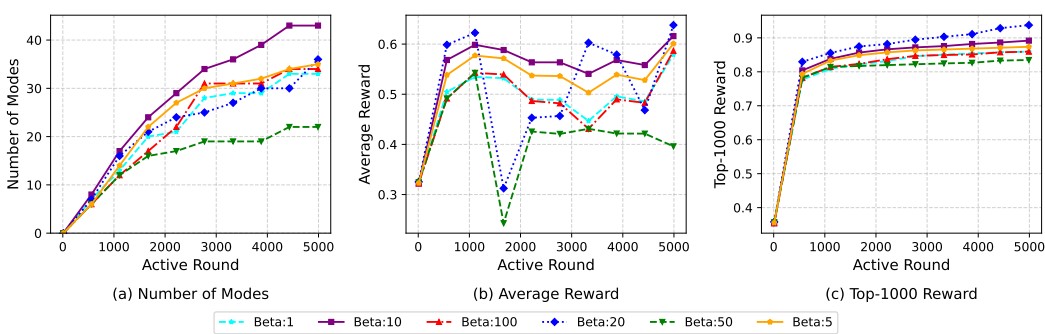

Figure 12: **Experimental result on RNA-Binding Task 3 with different $\beta$.**

Table 8: Comparison of Different Methods in the molecule generation task.

| Method | Modes $R > 7.5$ | Modes $R > 8$ | Top-1000 Reward | Top-1000 Similarity |
|---|---|---|---|---|
| Hard-Pruning | 1069 | 132 | 7.95 | 0.47 |
| Proportional | 7554 | 1190 | 8.19 | 0.46 |
| Random | 2179 | 284 | 8.03 | 0.47 |
| CBFlowNet | 12089 | 2952 | 8.31 | 0.49 |

When employing a simple hard-pruning approach without the CMAB framework, GFlowNet initially discovers numerous high-reward modes quickly. However, the mode distribution within this constrained subspace is sparse, and after these easily accessible modes are found, the discovery rate drops significantly as the remaining modes become increasingly difficult to identify.

In contrast, a random selection strategy explores the space uniformly by choosing super arms indiscriminately, yielding an expected reward equal to the average across all sub-spaces. This approach achieves performance comparable to the Trajectory Balance (TB) method.

A proportional selection strategy, which chooses arms according to their estimated rewards, naturally outperforms random selection by favoring higher-reward regions. However, it remains less aggressive than the top-K approach used in CBFlowNet. The proportional method discovers fewer total modes (7,554 vs. CBFlowNet's 12,089) yet achieves marginally better performance in top-1000 similarity metrics. The results suggest that while proportional selection maintains better diversity, CBFlowNet's more aggressive top-$K$ strategy enables superior overall mode coverage.

# I ILLUSTRATION OF WORKFLOW

Here, we present a case study to illustrate how the proposed method work in the bit sequence generation task. The set $M$ is constructed by randomly combining substrings derived from the base patterns $00000000, 11111111, 11110000, 00001111, 00111100$. For $K = 4$, theoretical analysis reveals that the optimal super arm configuration is $S = 0000, 1111, 1100, 0011$, which achieves perfect mode identification within its substate space with an average distance of $0.0$ to all target modes.

Initially, the base arm $0000$ gradually gains higher values (as shown in Figure 4), leading to a suboptimal super arm configuration $S = 0000, 1111, 1100, 0001$. The UCB mechanism in line 11 of Algorithm 1 then identifies $0011$ as a promising alternative - while its current estimated value is slightly lower than $0001$, its higher uncertainty (due to insufficient exploration) suggests significant potential. This triggers an exploration phase where the algorithm selects $S = 0000, 1111, 1100, 0011$ as the new super arm. Subsequent evaluations confirm that $0011$ consistently generates higher-quality candidates, and the value update in line 15 of Algorithm 1 reinforces its estimate in future rounds.

Notably, even though we know a priori that $S$ is optimal, the CMAB framework continues to explore alternative subspaces with some probability. This characteristic ensures the algorithm maintains the

capability to discover potentially better configurations while predominantly exploiting the known optimal solution, effectively balancing the exploration-exploitation trade-off throughout the learning process.

## J    DIVERSITY METRIC FOR MOLECULE GENERATION

While our current approach uses Tanimoto similarity with a 0.7 threshold for mode differentiation, it is still important to evaluate the mean internal Tanimoto similarity. Specifically, we compared the mean internal Tanimoto similarity among top-scoring candidates between CBFlowNet and TB. Given that increased top-$K$ reward typically correlates with higher molecular similarity (as demonstrated in (Malkin et al., 2022a)), we conducted additional comparative analyses between models with comparable reward performance: specifically, CBFlowNet at training rounds $10^4$ versus TB at rounds $10^5$. The results are summarized below:

Table 9: Performance comparison between differe nt methods.

| Method | Training Round | Top-100 | | Top-1000 | |
|---|---|---|---|---|---|
| | | Reward | Similarity | Reward | Similarity |
| TB | $10^5$ | 8.23 | 0.50 | 8.01 | 0.47 |
| CBFlowNet | $10^5$ | 8.43 | 0.55 | 8.31 | 0.49 |
| CBFlowNet | $10^4$ | 8.30 | 0.48 | 8.10 | 0.46 |

If we always sample through the same subspace, there is no doubt that the Tanimoto similarity of generated candidates is higher compared to sampling through the whole state space. However, the CMAB framework frequently changes the subspaces (as in line 11 of Algorithm 1) and the candidates sampled from different sub state spaces are more likely to have lower Tanimoto similarity, which alleviates this problem. This analysis demonstrates that while our approach may show slightly higher similarity within concentrated sampling periods, the overall diversity is maintained and even improved when considering equivalent performance levels, benefiting from our strategic subspace selection mechanism.

When comparing models with the same number of training rounds ($10^5$), CBFlowNet shows slightly higher Tanimoto similarity among top candidates. When comparing models with comparable reward performance (e.g., CBFlowNet at rounds $10^4$ vs TB at $10^5$ rounds), CBFlowNet achieves better diversity (lower similarity scores).

## K    TIME AND GPU MEMORY CONSUMPTION

Table 10 reports the runtime and GPU memory usage of TB-GFN and CBFlowNet in the molecule generation task. With the current implementation, training CBFlowNet for 100 rounds requires slightly more time than TB-GFN while consuming a comparable amount of GPU memory. Given the faster convergence of CBFlowNet, this marginal overhead does not substantially increase the overall training cost. Moreover, the extra sampling step can be efficiently parallelized using multi-threading, which further alleviates time consumption.

Table 10: Efficiency comparison between methods.

| Method | Time Cost (s / 100 rounds) | GPU Memory Cost |
|---|---|---|
| TB-GFN | 24.37 | 4827 MiB |
| CBFlowNet | 27.95 | 4840 MiB |

The time cost represents the duration required for completing 100 training rounds, where each round involves sampling candidate solutions for training and generating new candidates with a batch size of four. Thus, 100 rounds yield 400 candidates for evaluation. Multi-threading was employed to reduce time consumption.

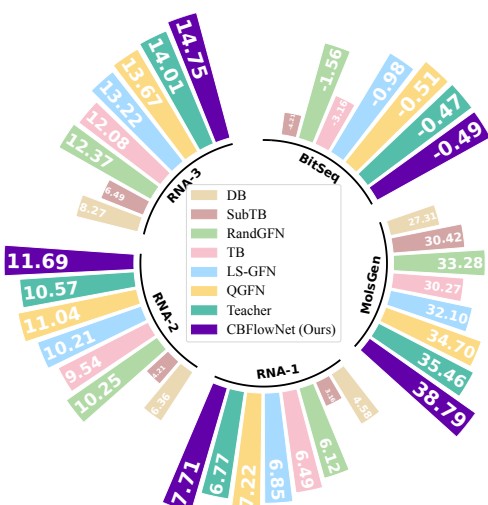

Figure 13: Experiment results on ELBO.

## L  EVIDENCE LOWER BOUND (ELBO)

We evaluate the goodness of fit to the target distribution using the evidence lower bound (ELBO) introduced by Kim et al. (2024c). ELBO is estimated by sampling $M$ candidates and averaging the estimated $\log Z$ through a transformed TB objective:

$$Z \prod_{t=1}^{n} P_F(s_t|s_{t-1}) = F(X) \prod_{t=1}^{n} P_B(s_{t-1}|s_t). \tag{11}$$

The corresponding ELBO is approximated as:

$$ELBO \approx \frac{1}{M} \sum_{i=1}^{M} \left( \log R(x_i) + \sum_{t=1}^{n_i} P_B(s_{t-1}|s_t) - \sum_{t=1}^{n_i} P_F(s_t|s_{t-1}) \right), \tag{12}$$

where $n_i$ is the length of the trajectory that generates terminal state $x_i$. Results are presented in Figure 13. The proposed CBFlowNet achieves slightly better performance than the baselines, with Teacher remaining the strongest competitor.

## M  SCALABILITY

We show that the proposed method scales effectively to larger action spaces both theoretically and empirically. When selecting $K$ arms from $N$ total base arms to form a super arm, the accuracy of reward estimates $\hat{\mu}_i$ typically requires more rounds to converge as $N$ grows. However, in practice, $K$ often scales proportionally with $N$—for example, setting $K = 0.1N$ (selecting $10\%$ of base arms). Under this scheme, the estimation accuracy of $\hat{\mu}_i$ remains stable with respect to $N$, leading to a fixed convergence rate.

Moreover, the computation of rewards $\hat{\mu}_i$ can be embedded within the flow-matching updates at negligible cost. Since our algorithm is heuristic, the additional computational burden is minimal, as also indicated in Table 10.

To further validate scalability, we experimented with enlarged action spaces. The molecule design task originally contains 105 building blocks with several stems each. By combining two actions into one base arm, the action space increases to $105 \times 105 = 11025$, denoted as CBFlowNet (CA). As shown in Table 11, CBFlowNet (CA) exhibits comparable or slightly better performance in discovering high-reward modes while incurring only negligible computational overhead.

Table 11: Performance comparison of different methods.

| Method | Training Round | Modes $R > 7.5$ | Modes $R > 8$ | Top-1000 Reward | Top-1000 Similarity | Time (s / 100 rounds) |
|---|---|---|---|---|---|---|
| TB-GFN | $10^5$ | 1915 | 233 | 8.01 | 0.47 | 24.37 |
| CBFlowNet | $10^5$ | 12089 | 2952 | 8.31 | 0.49 | 27.95 |
| CBFlowNet (CA) | $10^5$ | 12433 | 2520 | 8.29 | 0.50 | 28.73 |

## N  EXPERIMENTS WITH LARGE LANGUAGE MODEL TASK

We also report experiments on a task with dynamic reward distributions (see Section 6) as a limitation study. Following Hu et al. (2023), we considered a subjectivity classification task where each movie review is labeled as either objective or subjective. This task is particularly challenging, as it involves both the E-step and M-step of the EM algorithm, with GFlowNet serving as the inference model in the E-step.

We adopt the default settings from the public implementation of Hu et al. (2023). For fine-tuning, we tested both GPT-2 and GPT-J 6B backbones.

Table 12: Comparison of CBFlowNet and GFlowNet under GPT-2 and GPT-J backbones with different sample sizes. Results are reported as mean $\pm$ standard deviation.

| Method | GPT-2 | | | GPT-J 6B | | |
|---|---|---|---|---|---|---|
| | 10 Samples | 20 Samples | 50 Samples | 10 Samples | 20 Samples | 50 Samples |
| CBFlowNet | 0.59 $\pm_{0.02}$ | 0.63 $\pm_{0.03}$ | 0.78 $\pm_{0.02}$ | 0.71 $\pm_{0.02}$ | 0.83 $\pm_{0.01}$ | 0.90 $\pm_{0.01}$ |
| GFlowNet | 0.58 $\pm_{0.03}$ | 0.61 $\pm_{0.02}$ | 0.75 $\pm_{0.03}$ | 0.71 $\pm_{0.02}$ | 0.81 $\pm_{0.02}$ | 0.87 $\pm_{0.02}$ |

The results in Table 12 show that CBFlowNet marginally outperforms GFlowNet in test accuracy across all training sample sizes. However, this task highlights a limitation of CBFlowNet. The reward of a terminal state $Z$, defined as $p_{LM}(Z, Y \mid X)$, depends on both the label $Y$ and input $X$. For instance, the word *factual* may yield a high reward when the label is "objective" but a low reward when the label is "subjective." Thus, the high-reward state space shifts during training, which differs fundamentally from our original setting.

We categorize base arms into four groups: A) high-scoring under "objective" and low-scoring under "subjective"; B) high-scoring under "subjective" and low-scoring under "objective"; C) consistently high-scoring; D) consistently low-scoring.

Our framework is primarily designed to identify and filter arms of type D while retaining type C. In this dynamic setting, where types A and B fluctuate, we increased the base-arm set size $K$ to ensure sufficient coverage of relevant arms (types A, B, and D).

## O  ALGORITHM OF THE PROPOSED METHOD

Algorithm 1 presents the proposed Combinatorial Upper Confidence Bound (CUCB) framework augmented with a flow network. The algorithm alternates between an initialization phase, where all base arms are explored, and a main learning phase, where super arms are adaptively selected based on UCB values.

**Algorithm 1** Combinatorial Upper Confidence Bound (CUCB) with Flow Network

---

**Maintain for each arm** $i$:       • $T_i$: Number of times arm $i$ has been selected

      • $\hat{\mu}_i$: Empirical mean reward of arm $i$

**Input parameters:**       • $m$: Total number of base arms

      • $K = |\mathbb{S}|$: Size of super arm

1: **// Initialization Phase:**
2: **while** $\exists i \in \mathbb{S}$ with $T_i = 0$ **do**
3:      GFLOWNET.TRAIN($ALL$)
4:      Receive feedback from GFLOWNET.GEN($ALL$)
5:      Update $T_i$ and $\hat{\mu}_i$ for all $i \in \mathbb{S}$
6:      $m \leftarrow m + 1$
7: **end while**
8: **// Main Learning Phase:**
9: **for** $t = m$ **to** $T$ **do**
10:      Compute UCB for each arm:
11:        $\overline{\mu}_i \leftarrow \hat{\mu}_i + \sqrt{\frac{3 \ln t}{2 T_i}}$
12:      Select super arm $\mathbb{S} = \text{topK}(\overline{\mu}_1, \ldots, \overline{\mu}_m)$
13:      GFLOWNET.TRAIN($\mathbb{S}$)
14:      Receive feedback from GFLOWNET.GEN($ALL$)
15:      Update $T_i$ and $\hat{\mu}_i$ for all $i \in \mathbb{S}$
16: **end for**

---

