# OpenReview forum: "CBFlownet: Generating Higher-Quality Candidates via Combinatorial Bandits"
_ICLR.cc/2026/Conference — ICLR 2026 Conference Withdrawn Submission_

### Official Review · Reviewer_HxU9 · 2025-10-27

**Soundness:** 3
**Presentation:** 3
**Contribution:** 1
**Rating:** 2
**Confidence:** 3

**Summary:**

This paper introduces CBFlowNet, a method to overcome the problem of oversampling the low-reward regions and undersampling the high reward regions in GflowNets.

Methodology:
To do so, they prune the actions that yield to terminal states with a low expected reward.
They root their method in the framework of combinatorial multi-arms bandit: form a state , given N possible actions (i.e. arms ), they choose the top-K arms (K < N ) according to a score (defined below). The other actions are pruned.
The score for arm i at training step t is simply the combination of:
    - the mean reward of terminal states that are sampled at time t from arm i  (this is not specified in the paper, but I guess the mean reward is obtained with on-policy sampling), which encourages exploitation,
    - and a term rooted in UCB literature and that increases the score for rarely seen arms (i.e. encourages exploration).

Experiments:
They run experiments on the bit-sequence task (with a modified and more complex DAG, because the original DAG is a binary tree and pruning an action there means we never visit all the subtree which comes after that action), the molecule design task and the RNA task.
They compare to multiple other baselines () .
They compare the different methods in terms of : cumulative number of discovered modes, reward of the top-X samples, average reward.  They show that CBFLowNet outperforms other methods, but do not provide a theoretical justification of why it is the case.


Contributions:
- Adding the UCB term to the score of each arm
- Converting the GFN-binary trees, which do not yield themselves to action pruning (because this directly removes a full subtree), to non-binary trees

**Strengths:**

I like how the experiments section is presented:
- Extensive set of experiments on 3 different environments (same as those in the original GflowNet paper)
- Nice visualisation of the evolution of the action values
- Comparing to multiple baselines to train GflowNets with the goal of doing better credit assignment (LS , QGFN, student-teacher)
- Adding the baseline of RandGFN

**Weaknesses:**

- Except for the added UCB term ( sqrt ( ln(t) / T_i ) )  that encourages exploration, it is not clear what is the added contribution of this work compared to the previous work of Lau et al. It seems to the reviewer that here, the "mean on-policy reward" of each arm is the same as the Q value function of each arm (because Q(s,a) =  Expectation_{ \tau = (s, .. s_f) \sim P_F(.| s, a )}  (  Rew(s_f)). The only difference is that in this work, the value function is not a learned neural net, whereas in Lau et al,  Q was parametrized with a neural network, allowing them to generalise to unseen states. Also, it is claimed in the paper that they introduce pruning and that is was not used in Lau et al, however it has been used there, wee eqs 4 and 5 in https://arxiv.org/pdf/2402.05234
- The maths and notations is not very rigorous sometimes or not defined
   - e.g. Formally, if arm = arm′ , then Xt =d X′t must hold   . what is d ? Why is there a " ' "? Could you elaborate what you mean by this sentence?
   - Why are the properties of monononicity and smoothness are introduced?
   - monononicity:  the last sentence is not necessarily true: imagine a flow network that, for each base arm, puts all the flow in the  maximal-reward terminal state reachable from that arm, and zero flow for all other terminal states. However such flow loses the gflownet property of sampling proportionnally to the reward

**Questions:**

Primary questions:
- Could you explain the difference between your work and the work of Lau et al, except for the UCB added term? It seems to me that the core is the same (except that in your case the score is not learned with a neural net).
If I were to compare your work with using the Q value function as Lau et al. and adding to it the UCB term to it, what would be the difference? What is the contribution of your work compared to Lau et al?

- Why did you choose to root your work in combinatorial multi armed bandits? I don't see something that is truly combinatorial here, as there is no interaction between the different arms, and the score of each arm is independent of other arms. In the end, the score amounts to the top-K arms, so there is no need to do a search of the best subset of size K in a set of size N.

- Have you done an ablation study using only the mean empirical reward as a score, i.e. without the exploration term in the UCB?

Secondary questions (not as important):

- Could you elaborate why is independence among base-arm rewards violated under naive pruning?
- In eq.3 , why is it necessary to normalise the reward for effective exploration? how are the reward normalised if we don't have a normalisation constant?

---

> ### Author Response · Authors · 2025-11-22
>
> Dear Reviewer HxU9,
>
> Thank you very much for your valuable comments, which are crucial to the improvement of our paper. We would like to clarify your concerns point by point in the following.
>
> >W1. Except for the added UCB term that encourages exploration, it is not clear what is the added contribution of this work compared to the previous work of Lau et al. The only difference is that in this work, the value function is not a learned neural net, whereas in Lau et al, Q was parametrized with a neural network, allowing them to generalise to unseen states. Also, it is claimed in the paper that they introduce pruning and that is was not used in Lau et al, however it has been used there, wee eqs 4 and 5 in https://arxiv.org/pdf/2402.05234
>
>
> Thank you very much for your insightful comment. QGFN is a great work which inspires us a lot. But we would like to clarify that there exist key differences between our work and QGFN.
>
> QGFN prunes actions based on Q-value thresholds (p-quantile or a fraction of the max), which permanently masks portions of the state space. Although Q-values evolve during training, actions pruned early are rarely revisited or updated, even if they eventually become promising. This may lead to irreversible information loss, and as acknowledged in QGFN, "aggressive pruning can be harmful—especially when the action set is small".
> Additionally, once Q stabilizes, the pruning rule becomes fixed, reducing the ability to recover unexplored high-reward regions.
>
>
> In contrast, our method never prunes states or actions based on local Q-values. Instead, we partition the original search space into dynamic subspaces defined by super arms. These subspaces are actively **cycled** and **re-evaluated** under a CMAB framework, ensuring that **no region becomes permanently inaccessible**. This structural property avoids the core issue of irreversible pruning and preserves the ability to discover new high-reward areas even in later training phases. Our algorithm purposefully includes mechanisms (`Line 11 of algorithm 1`) that ensure the sampler does not stay in a single high-scoring subspace indefinitely. This guarantees continual exploration at the subspace level—a property that QGFN does not possess due to its fixed pruning rule.
>
> The key difference between our proposed method and previous works such as QGFN lies in a new problem formulation and exploration perspective. We proposed the concept of exploring multiple subspaces instead of operating on the whole state space, which is the core idea behind our proposed method. We view GFlowNet sampling as a process of **searching for high-scoring subspaces** within a vast collection of potential subspaces, whereas prior works primarily operate from the perspective of **the entire state space**. In consequence, our method pays attention to both exploitation and exploration while other methods only focus on exploitation, with information loss and inferior performance. Our approach thus introduces a **new angle** for improving GFlowNet performance.

---

> > ### Author Response · Authors · 2025-11-22
> >
> > >Q2. Why did you choose to root your work in combinatorial multi armed bandits? I don't see something that is truly combinatorial here, as there is no interaction between the different arms, and the score of each arm is independent of other arms. In the end, the score amounts to the top-K arms, so there is no need to do a search of the best subset of size K in a set of size N.
> >
> > Thanks for raising this question. The combinatorial multi-armed bandit (CMAB) framework indeed assumes that **base-arm rewards are independent**, and this is not a limitation but a foundational premise that enables tractable reasoning over subsets of arms. Our setting aligns naturally with this assumption. While it may appear that “the score only depends on the top-K arms,” identifying the top-K combination in a large candidate pool is inherently a combinatorial optimization problem, especially in high-dimensional GFlowNet environments with exponentially large state spaces. Efficiently navigating this combinatorial structure is precisely what CMAB methods are designed for. More importantly, our objective is not merely to recover the top-K arms. Instead, our method deliberately emphasizes sampling across promising subspaces, guided by both reward and exploration signals. This design offers two key benefits: it increases sample diversity, since candidates from different subspaces tend to be less similar, and it accelerates mode discovery, as it is easier to find previously undiscovered modes in underexplored regions compared to well-explored areas where accessible modes have already been identified. This concept of exploring multiple subspaces is, in fact, the **core idea** behind our proposed method.
> >
> >
> > >Q3. Have you done an ablation study using only the mean empirical reward as a score, i.e. without the exploration term in the UCB?
> >
> > Thank you for pointing this out. We realized that the ablation you referred to: the variant that uses only the empirical mean reward as the scoring function (removing the exploration term from UCB) was not included in the previous version. To address this, we have now conducted the requested experiment and report the results in the table below(original table with different strategies can be found in `Table 8` of our paper):
> >
> > | Method        | Modes R>7.5 | Modes R>8 | Top-1000 Reward | Top-1000 Similarity |
> > |---------------|-------------|-----------|------------------|----------------------|
> > | Hard-Pruning  | 1069        | 132       | 7.95             | 0.47                 |
> > | Proportional  | 7554        | 1190      | 8.19             | 0.46                 |
> > | Random        | 2179        | 284       | 8.03             | 0.47                 |
> > | CBFlowNet     | 12089       | 2952      | 8.31             | 0.49                 |
> > | No-exploration     | 2889       | 543      | 8.13             | 0.51                 |
> >
> > Removing the exploration term indeed allows the model to pick up several high-reward trajectories early (“No-exploration”), but the discovery rate quickly **drop** because the policy becomes **over-concentrated** in the initially visited subspace and fails to **explore** alternative regions. As the table shows, both the total number of discovered modes and the number of high-reward modes remain far below those achieved by CBFlowNet. This confirms that the exploration component in our CMAB-guided pruning is crucial for sustained mode discovery rather than early greedy improvements.
> >
> >
> > >Q4. Could you elaborate why is independence among base-arm rewards violated under naive pruning?
> >
> > See W2-1.
> >
> > >Q5. In eq.3 , why is it necessary to normalise the reward for effective exploration? how are the reward normalised if we don't have a normalisation constant?
> >
> > The normalization step is introduced to scale the $\hat{\mu_i}$ so that the exploration term in `Equation 4` functions properly. Without normalization, rewards across different tasks may have vastly different magnitudes, which can skew the balance between exploitation and exploration. By normalizing, we ensure that the exploration bonus contributes meaningfully and comparably across all candidates, allowing the algorithm to explore under-sampled regions effectively.
> >
> > As for the normalization constant, actually, most experiments already provided the scaled reward function themselves. For instance, in molecule generation task, the reward normalization is already done by the official code of GFlowNet(https://github.com/GFNOrg/gflownet). Even if some experiments lack an explicit normalization constant, we can still use relative normalization methods such as min-max scaling within the sampled candidates so far or running statistics (e.g., moving mean and variance) to dynamically adjust the scale. This avoids the need for a fixed global constant while still preserving consistent and stable exploration behavior throughout training.

---

> ### Author Response · Authors · 2025-11-22
>
> >W2-1. The maths and notations is not very rigorous sometimes or not defined
> e.g. Formally, if arm = arm′ , then Xt =d X′t must hold . what is d ? Why is there a " ' "? Could you elaborate what you mean by this sentence?
>
> Thanks for pointing out this issue. We appreciate the opportunity to clarify the notation. In our setting, we consider two super-arm sets, denoted as ${arm_1, \dots, arm_K}$ and ${arm_1', \dots, arm_K'}$. If the $i$-th base arm in the first set is the same as the $j$-th base arm in the second set, i.e., $arm_i = arm_j'$, then their corresponding reward distributions should also be identical. Formally, we denote this as $X_i^t \stackrel{d}{=} X_j'^t$, where $\stackrel{d}{=}$ indicates equality in distribution.
>
> Our point is that if the GFlowNet is constrained to sample only within a subspace defined by a particular super arm, then the expected reward of each base arm becomes dependent on which super arm it belongs to. In such cases, even intrinsically poor base arms may appear to have high rewards simply because they are grouped within a “good” super arm. This violates the independence assumption on base arms, which is a key prerequisite for applying CMAB algorithms.
>
> We will include a clearer explanation of these notations and assumptions in the revised version.
>
> >W2-2. Why are the properties of monononicity and smoothness are introduced?
>
> Thanks for the question. Monononicity and smoothness are two standard and mild assumptions in CMAB framework, without which the whole problem will become intractable. Specifically, monotonicity ensures that adding more base arms with higher individual rewards does not decrease the expected overall reward, preserving a consistent reward structure across combinations. In GFlowNet scenario, monotonicity ensures that a better subspace(all base arms have higher rewards) is expected to achieve higher rewards than a bad subspace, neglecting sampling noise. Smoothness, on the other hand, bounds the sensitivity of the global reward to changes in individual base-arm rewards, ensuring that small variations in arm-level outcomes do not cause disproportionately large changes in the combinatorial reward. In GFlowNet scenario, since the global reward of a super arm is defined as the summation of base arms, this assumption natually holds.
>
>
> >W2-3. monononicity: the last sentence is not necessarily true: imagine a flow network that, for each base arm, puts all the flow in the maximal-reward terminal state reachable from that arm, and zero flow for all other terminal states. However such flow loses the gflownet property of sampling proportionnally to the reward
>
> This assumption assumes that if a flow network has the gflownet property then it should has monononicity, rather than the reverse way. As you mentioned, the flow network given above loses the gflownet property of sampling proportionnally to the reward. As we are optimizing the sampling process of GFlowNets and leave GFlowNet training objective unchanged, the flow network you mentioned is impractical and should not exist.
> Under the right proportionally flow structure, monononicity always holds.
>
> >Q1. Could you explain the difference between your work and the work of Lau et al, except for the UCB added term? It seems to me that the core is the same (except that in your case the score is not learned with a neural net). If I were to compare your work with using the Q value function as Lau et al. and adding to it the UCB term to it, what would be the difference? What is the contribution of your work compared to Lau et al?
>
> See W1.

---

### Official Review · Reviewer_pnhU · 2025-10-29

**Soundness:** 3
**Presentation:** 3
**Contribution:** 3
**Rating:** 6
**Confidence:** 4

**Summary:**

The paper proposes CBFlowNet, which integrates the CMAB framework to prune the DAG that leads to proper exploration. Authors try to systematically integrate CMAB into the GFlowNet sampling process. Experiment results validate that the proposed method outperforms prior GFlowNet baselines.

**Strengths:**

- Over-exploration is one of the prevalent challenges when utilizing GFlowNets in real-world settings [1]. The paper seeks to address this crucial challenge using CMAB framework, a powerful tool for extracting a useful subset of actions.

- Experiment results validate the effectiveness of the proposed method. Especially, it shows a large improvement in the molecule design task, where the action space is much larger than in other benchmarks, making appropriate pruning effective.]


[1] Kim, Minsu, et al. "Local Search GFlowNets." The Twelfth International Conference on Learning Representations.

**Weaknesses:**

- In bit sequence generation, we can modify the $k$ to study the effect of the trade-off between trajectory lengths and the action space sizes.  With a single experiment, I'm not sure that the proposed method works well with different environment configurations.

- Similar to above, there are biological sequence tasks like GFP and AAV, which contain many more state-independent actions (20 amino acids). It would be nice to add some analysis of whether the proposed method works in those settings.

**Questions:**

Here are some comments I want to suggest:
- In the introduction, there is a section where training GFlowNets with tempered distribution. It would be nice to add Temp-GFN [1], which tries to learn from the diverse tempered distribution for improving sampling process.

- In Figure 3, some symbols are crashed.

- In the experiment section, it is better to write like "Trajectory Balance (TB, Malkin et al., 2022a)" rather than "Trajectory Balance(TB)(Malkin et al., 2022a)".

- I cannot find which training objective is used (e.g., FM, DB, TB) for CBFlowNet. Maybe I missed it but it would be nice to clearly indicate that part. I'm also curious about the proposed method can be seamlessly applied to different training strategies.

[1] Kim, Minsu, et al. "Learning to Scale Logits for Temperature-Conditional GFlowNets." Forty-first International Conference on Machine Learning.

---

> ### Author Response · Authors · 2025-11-22
>
> Dear Reviewer pnhU,
>
> Thank you very much for your valuable comments, which are crucial to improve of our paper. We would like to address your concerns point by point.
>
> > W1. In bit sequence generation, we can modify the to study the effect of the trade-off between trajectory lengths and the action space sizes. With a single experiment, I'm not sure that the proposed method works well with different environment configurations.
>
> Apart from the bit sequence task, we also evaluated the effect of varying trajectory lengths in the molecular design experiment. As shown in `Appendix M` and `Table 11`, we combined the molecular design blocks so that the size of the base-arm set increased from 105 to 105×105. The results indicate that this operation can even lead to improvements on certain metrics. However, because the original base-arm set was already sufficiently well-balanced, the overall improvements were not significant. Therefore, we recommend applying this additional combination operation only when the width and length of the flow network are extremely imbalanced.
>
>
> > W2. Similar to above, there are biological sequence tasks like GFP and AAV, which contain many more state-independent actions (20 amino acids). It would be nice to add some analysis of whether the proposed method works in those settings.
>
> Thank you for the constructive suggestion. We additionally conducted experiments on the AMP task, which contains 20 state-independent actions with the action set \{'A', 'C', 'D', 'E', 'F', 'G', 'H', 'I', 'K', 'L', 'M', 'N', 'P', 'Q', 'R', 'S', 'T', 'V', 'W', 'Y'\}. Our method achieved substantial improvements across all evaluation metrics on this task, further demonstrating its effectiveness in settings with many state-independent actions.
>
> | Method    | Top-100 Performance     | Top-100 Diversity      | Top-1000 Performance    | Top-1000 Diversity     |
> |-----------|--------------------------|-------------------------|--------------------------|-------------------------|
> | CBFlowNet | 0.9549 ± 0.0031          | 26.79 ± 0.42            | 0.9496 ± 0.0028          | 28.10 ± 0.55            |
> | TB-GFN    | 0.9316 ± 0.0045          | 22.31 ± 0.38            | 0.8769 ± 0.0062          | 23.42 ± 0.44            |
>
> The results clearly show that CBFlowNet substantially outperforms the TB-GFN baseline in both performance and diversity for the top-100 and top-1000 generated sequences. Notably, even the Top-1000 Performance of CBFlowNet outperforms the Top-100 Performance of TB-GFN. Besides, it is worth noting that the rewards in this task are provided by a proxy model and therefore contain some degree of inaccuracy. Given this limitation, the performance achieved by our method is already approaching the upper bound imposed by the predictive capacity of the proxy. These findings demonstrate that our method remains effective even in environments with large state-independent action spaces, such as AMP and other biological sequence tasks like GFP and AAV. This supports the broader applicability of our method, confirming its potential for real-world biological design scenarios.
>
> > Q1. In the introduction, there is a section where training GFlowNets with tempered distribution. It would be nice to add Temp-GFN [1], which tries to learn from the diverse tempered distribution for improving sampling process.
>
> Thank you for your suggestion, we will add Temp-GFN as reference and analyze the difference between it and our work in revision.

---

> ### Author Response · Authors · 2025-11-22
>
> > Q2. In Figure 3, some symbols are crashed.
>
> This issue is caused by a rendering problem specific to the version of Microsoft Edge. We recommend opening the PDF with a dedicated PDF viewer or a browser with more reliable PDF rendering support. The symbols will display correctly in those environments.
>
> > Q3. In the experiment section, it is better to write like "Trajectory Balance (TB, Malkin et al., 2022a)" rather than "Trajectory Balance(TB)(Malkin et al., 2022a)".
>
> Thanks for your suggestions on formatting. We will follow your suggestion and reformat the citation in our revision.
>
>
> > Q4. I cannot find which training objective is used (e.g., FM, DB, TB) for CBFlowNet. Maybe I missed it but it would be nice to clearly indicate that part. I'm also curious about the proposed method can be seamlessly applied to different training strategies.
>
> Thanks for pointing this out. We use the Trajectory Balance (TB) objective as the default training objective for CBFlowNet. We will clarify this in the method section by adding an explicit description of the objective used.
> Moreover, since our method operates solely at the sampling stage and does not modify the training process, it can be seamlessly combined with any existing training objective as you mentioned. Our approach provides a general and flexible framework that is compatible with current objectives as well as potential future ones.

---

> > ### Comment · Reviewer_pnhU · 2025-11-26
> >
> > Thanks for the authors' rebuttal. While most of my concerns have been resolved, it would be better to show experimental results on bit sequence tasks by varying $k$. I think the results on AMP task is also nice, and I appreciate it, but it would be better to show experimental results on AAV and GFP tasks, which have much longer trajectory lengths. Anyway, I maintain the positive rating.

---

### Official Review · Reviewer_ne9Q · 2025-10-30

**Soundness:** 2
**Presentation:** 2
**Contribution:** 2
**Rating:** 2
**Confidence:** 3

**Summary:**

The paper proposes **CBFlowNet**, a hybrid framework that augments **Generative Flow Networks (GFNs)** with a **Combinatorial Multi-Armed Bandit (CMAB)** component. The goal is to **reduce over-exploration** in GFNs by **pruning low-quality actions** via CMAB, thereby **biasing the flow network toward high-reward regions** without sacrificing diversity. The authors instantiate this idea using the **CUCB (Combinatorial Upper Confidence Bound)** algorithm to dynamically select a **super arm** of promising actions, restrict GFN sampling to this subspace, and iteratively refine the selection based on observed rewards. Experiments are conducted on **bit sequence generation**, **molecule design**, and **RNA sequence design**, where CBFlowNet reportedly **discovers more high-reward modes** and **converges faster** than standard GFN baselines.

**Strengths:**

- **Originality**:
  The **combination of CMAB and GFNs** is **novel**. While GFNs have been enhanced with Q-values, local search, or adaptive teachers, **using CMAB to prune the action space** is **a new angle** that addresses the **over-exploration** issue in a **principled, bandit-driven** way.

- **Quality**:
  The paper is **technically sound**, with **careful algorithm design**, **theoretical justification** (monotonicity & bounded-smoothness assumptions), and **extensive ablations** (K, ε, β, alternative arm-selection rules). The **empirical gains** in **mode discovery** and **top-1000 reward** are **consistent** across tasks.

- **Clarity**:
  The **framework is well-structured**, with **clear notation**, **illustrative figures**, and **detailed pseudocode**. The **decomposition of actions** into **state-dependent** and **state-independent** components is **intuitive** and **generalizable**.

- **Significance**:
  If **scalable**, CBFlowNet could **broadly improve** **combinatorial generative modeling** in **drug discovery**, **sequence design**, and **program synthesis**, where **high-reward diversity** is **critical**.

**Weaknesses:**

### W1. **Motivation for CUCB is Weak**
- The paper **assumes** that **CUCB is the right bandit algorithm** without **justifying why**.
- **No comparison** with **other CMAB algorithms** (e.g., **CTS**, **C-KL-UCB**, **ESCB**, **Thompson Sampling**) or **non-bandit pruning** (e.g., **reinforcement learning with curiosity**, **evolutionary search**).
- **Missing ablation**: **Replace CUCB with random pruning** → **same gain?** If so, the **contribution is incremental**.

### W2. **Generality Beyond GFNs is Unexplored**
- The **core idea**—**prune actions via CUCB**—is **not validated** on **other generative frameworks**:
  - **Flow Matching** (no MDP)
  - **Soft Q-Learning** (no flow constraints)
  - **Diffusion Models** (no sequential decisions)
- **No theoretical argument** that **CUCB pruning** is **uniquely compatible** with **GFNs**.
  → **Limits scope**; readers **cannot tell** if this is **a GFN trick** or **a general principle**.

### W3. **Scalability of Combinatorial Action Pruning is Unaddressed**
- **Super-arm selection** is **top-K** from **N base arms**; **complexity is O(N log K)** per round.
- **N = 10⁵** in **molecule design**; **K = 30** → **~3×10⁶ arm-updates** per CUCB round.
- **No wall-clock breakdown** vs. **standard GFN**; **no memory footprint** as **N grows**.
- **Missing experiment**: **Increase N** (e.g., **10⁵ → 10⁶**) and **report**:
  - **Time per round**
  - **GPU memory**
  - **Regret degradation**



CBFlowNet offers a **creative bandit-based pruning mechanism**, and the **empirical gains are real**. However, the **contribution is incremental** because:

1. **No strong evidence** that **CUCB is critical**—**random pruning** might suffice.
2. **Generality** outside **GFNs** is **unexplored**; could be **a GFN-specific hack**.
3. **Scalability analysis** is **missing**; **10⁵ arms** with **K = 30** already feels **heavy**, and **no ceiling** is provided.

A **rebuttal** that **ablate CUCB vs. random pruning**, **tests Flow Matching**, and **scales N to 10⁶** could **flip** my decision.

**Questions:**

### Q1. Why CUCB?
> Can you **compare CBFlowNet** with **Thompson-Sampling CMAB** or **ESCB**?
> If **CUCB is replaced by random top-K pruning**, how much **performance drops**?
> This would **isolate** the **bandit contribution** from the **pruning contribution**.

### Q2. Does this work outside GFNs?
> Have you **tested CUCB pruning** on **Flow Matching** or **Soft Q-Learning** for **similar tasks**?
> If **no**, what **theoretical property** of **GFNs** makes **CUCB pruning essential**?

### Q3. Scalability ceiling?
> What is the **largest N** (base arms) you have **successfully run**?
> Provide **time & memory curves** for **N = 10³, 10⁴, 10⁵, 10⁶**.
> Is **K = 0.1N** still **tractable** when **N = 10⁶**?

### Q4. Dynamic rewards?
> Section 6 mentions **limitation** under **non-stationary rewards**.
> Could **non-stationary CMAB** (e.g., **Discounted UCB**, **Sliding-Window UCB**) **extend CBFlowNet** to **dynamic environments**?

### Q5. Baseline fairness?
> **TB-GFN** is **trained 5× longer** in **bit sequence** (50k → 10k steps).
> Did you **match compute budget** or **wall-clock time**?
> Please **report** **convergence curves vs. actual compute**.

---

> ### Author Response · Authors · 2025-11-22
>
> Dear Reviewer ne9Q,
> Thanks for your recognition of our novelty and clarity. Thank you very much for your valuable comments, which are crucial to the improvement of our paper. We would like to clarify your concerns point by point in the following.
>
> >W1-1. Motivation for CUCB is Weak. The paper assumes that CUCB is the right bandit algorithm without justifying why. No comparison with other CMAB algorithms (e.g., CTS, C-KL-UCB, ESCB, Thompson Sampling) or non-bandit pruning (e.g., reinforcement learning with curiosity, evolutionary search).
>
> The key insight of this work is to propose a **pruning framework**.
> It is important to clarify that our goal is not to identify the single best bandit algorithm. CUCB was chosen simply because it is a suitable and well-understood method that aligns with our motivation. However, the framework itself is **algorithm-agnostic**: other CMAB algorithms (e.g., CTS) can be incorporated as well, even though exploring all these variants is beyond the scope of this work.
>
> To further address this concern, we have conducted an additional experiment where we replaced CUCB with other CMAB algorithms, the results are listed below:
>
> | Method | Modes R>7.5 | Modes R>8 | Top-K Reward |
> |--------|-------------|-----------|--------------|
> | CUCB   | 13074       | 3207      | 8.436        |
> | CTS    | 12220       | 2725      | 8.431        |
> | ESCB   | 13568       | 3315      | 8.442        |
>
> These results confirm that our framework is not dependent on CUCB. Replacing CUCB with CTS or ESCB yields very similar performance, with ESCB even slightly outperforming CUCB in all metrics. This demonstrates that the improvement comes from the pruning framework itself, not from a particular bandit choice. CUCB is simply one reasonable instantiation, and the framework remains effective across different CMAB algorithms.
>
> Regarding non-bandit pruning strategies such as curiosity-driven reinforcement learning or evolutionary search, we note that while related, they involve substantially different design challenges, including well-known issues such as curiosity traps and detachment. These directions fall outside the focus of this paper, which centers specifically on bandit-based pruning within our proposed framework.
>
> >W1-2. Missing ablation: Replace CUCB with random pruning → same gain? If so, the contribution is incremental.
>
> We actually had experiment of replacing CUCB with random pruning in our manuscript. As shown in `line 306`, we introduced a baseline called **RandGFN** and made comprehensive comparisons with all baselines across all tasks. In addition, we explored more pruning strategies, which are reported and discussed in `Appendix H`.
>
> Importantly, replacing CUCB with random pruning leads to significantly worse performance. This demonstrates that our improvement is not incremental, i.e., the choice of CUCB is essential for achieving the reported gains.
>
> >W2. Generality Beyond GFNs is Unexplored. The core idea—prune actions via CUCB—is not validated on other generative frameworks like Flow Matching, Soft Q-Learning and Diffusion Models. No theoretical argument that CUCB pruning is uniquely compatible with GFNs. Limits scope; readers cannot tell if this is a GFN trick or a general principle.
> >
>
> Although both GFlowNets and frameworks like Flow Matching, Soft Q-Learning and Diffusion Models are generative models, their definition on **generative** are quite different. GFlowNets define generation as constructing an object step-by-step through a sequence of actions, aiming to sample **discrete** structures in proportion to their reward. In contrast, Flow Matching and Diffusion Models define generation as transforming noise into data by learning continuous flows or denoising processes from **large offline datasets with cotinuous space**. Thus, GFlowNets are generative in an **online, reward-driven** sense, whereas FM, Diffusion, and SQLearn are generative in an **offline, data-driven** sense.
>
> While our method is a general principle, it can only operate in frameworks with certain structural properties. CUCB-based pruning requires a **discrete** action space, which rules out **continuous** space generators like Flow Matching and Diffusion Models. It also relies on **online sampling**, this is incompatible with offline-trained models which do not interact with the environments at all, like Flow Matching and Diffusion Models. Finally, the underlying generative framework must aim to produce diverse high-reward candidates rather than seeking for a single optimum. Traditional RL methods optimize for a maximum-return policy and thus using pruning strategies in our proposed method does not bring clear benefits.
>
> Under these characteristics, GFlowNets naturally satisfy all required conditions, while Flow Matching, Diffusion Models, and Soft Q-Learning fundamentally lack the structural elements needed for CUCB-based pruning to be meaningful or effective.

---

> ### Author Response · Authors · 2025-11-22
>
> >W3. Scalability of Combinatorial Action Pruning is Unaddressed.Super-arm selection is top-K from N base arms; complexity is O(N log K) per round. N = 10⁵ in molecule design; K = 30 → ~3×10⁶ arm-updates per CUCB round.No wall-clock breakdown vs. standard GFN; no memory footprint as N grows. Missing experiment: Increase N (e.g., 10⁵ → 10⁶) and report:Time per round,GPU memory,Regret degradation
>
> In implementation level, the updation is not conducted each training round but through a fixed interval(e.g. 1000 rounds per updation) to reduce variance and stablize the results. Assuming we set the interval as I, the complexity is $O(\frac{N log K}{I})$. In implementation level, we use **torch.topK()** to select the top-K high-scoring base arms, which can be accelerated by GPU. Therefore, in most cases, the time consumption is minor compared to the time spent training neural netowrks.
> To further address your concerns, we conduct an additional study including the wall-clock break down and the memory spent with several different base arm size($10^2, 10^4, 10^6$) in molecule generation tasks.
>
> | Method            | Sampling Time | Training Time | Update Time | Other Time | GPU Memory | CPU Memory |
> |------------------|---------------|----------------|--------------|-------------|------------|------------|
> | CBFlowNet (1e2)  | 97.07s        | 308.90s        | 0.03s        | 151.9s      | 4125.0MB   | 2226.9MB   |
> | CBFlowNet (1e4)  | 98.69s        | 311.23s        | 0.15s        | 153.1s      | 4148.3MB   | 2251.2MB   |
> | CBFlowNet (1e6)  | 99.61s        | 313.61s        | 0.66s        | 155.7s      | 4164.5MB   | 2301.7MB   |
> | GFlowNet         | 50.70s        | 310.64s        | 0.00s        | 153.7s      | 4112.4MB   | 2217.3MB   |
>
>
> >Q1. Why CUCB? Can you compare CBFlowNet with Thompson-Sampling CMAB or ESCB? If CUCB is replaced by random top-K pruning, how much performance drops? This would isolate the bandit contribution from the pruning contribution.
>
> Same as W1.
>
> >Q2. Does this work outside GFNs? Have you tested CUCB pruning on Flow Matching or Soft Q-Learning for similar tasks? If no, what theoretical property of GFNs makes CUCB pruning essential?
>
> Same as W2.
>
> >Q3. Scalability ceiling? What is the largest N (base arms) you have successfully run? Provide time & memory curves for N = 10³, 10⁴, 10⁵, 10⁶. Is K = 0.1N still tractable when N = 10⁶?
>
> Same as W3. The largest number of base arms we have successfully run is $10^6$. In W3, we report the wall-clock breakdown for different N. And even for K = 0.1N, the time spent at updating base arms is still minor compared to the time spent to neural network training.

---

> ### Author Response · Authors · 2025-11-22
>
> >Q4. Dynamic rewards? Section 6 mentions limitation under non-stationary rewards. Could non-stationary CMAB (e.g., Discounted UCB, Sliding-Window UCB) extend CBFlowNet to dynamic environments?
>
> The notion of dynamic rewards in our context is fundamentally different, and even non-stationary CMAB algorithms cannot handle such environments. The reason is straightforward: in tasks like subjectivity classification, the reward of a terminal state $Z$, defined as $p_{LM}(Z, Y \mid X)$, depends on both the label $Y$ and input $X$. For instance, the word **factual** may yield a high reward when the label is **objective** but a low reward when the label is **subjective**. The data may shift abruptly—for example, switching suddenly from predominantly subjective labels to predominantly objective ones. In such cases, even methods based on sliding windows or discounting cannot adapt effectively, because they still rely on the assumption of smooth temporal evolution, which does not hold here. The details can be found in `Appendix N`.
>
>
> >Q5. Baseline fairness? TB-GFN is trained 5× longer in bit sequence (50k → 10k steps). Did you match compute budget or wall-clock time? Please report convergence curves vs. actual compute.
>
> The baseline comparison is fair, because our method also uses the trajectory-balance objective as the default objective. Considering that our proposed method already discovered all modes within 10k steps, we do not follow the same training steps setting.
> To further address your concern, we conduct an additional experiment where TB-GFN is trained 50k steps, and compare the corresponding performance as well as the compute budget and wall-clock time.
>
> | Method    | Active Round | Discovered Modes | Total  Time   |
> |-----------|---------|--------|----------|
> | CBFlowNet | 10k     | 60     |  3867 s  |
> | TB-GFN    | 10k     | 16     |  3504 s  |
> | TB-GFN    | 50k     | 41     | 19072 s  |
>
> Our additional analysis demonstrates that the baseline comparison is indeed fair.
> Under the same compute budget (10k steps, similar wall-clock time), TB-GFN discovers only 16 modes, whereas CBFlowNet discovers all 60 modes. This confirms that the improvement does not come from longer training, but from the proposed sampling strategy.
>
> Furthermore, even when TB-GFN is trained 5× longer (50k steps)—resulting in 5.4× more wall-clock time—it still discovers only 41 modes, significantly fewer than the 60 modes achieved by CBFlowNet within only 10k steps. This clearly indicates that CBFlowNet is not only more sample-efficient, but also more compute-efficient, discovering modes 7.8× faster per unit time.
>
> Therefore, our method provides substantial improvements even under strictly matched compute budgets, and the advantage persists—and even grows—when the baseline is allowed to train much longer.

---

### Official Review · Reviewer_9Be3 · 2025-10-31

**Soundness:** 1
**Presentation:** 2
**Contribution:** 3
**Rating:** 2
**Confidence:** 3

**Summary:**

The paper proposes a strategy for GFlowNet search space pruning to accelerate mode discovery.

**Strengths:**

- The authors propose a very interesting approach for GFlowNet search space pruning.
- Theoretical analysis is provided for selected components of the method.
- The authors report performance improvement (having said that, I have some questions about the experiments below).

**Weaknesses:**

- The authors claim that “Optimizing the inverse temperature parameter β presents non-trivial challenges, as its selection critically impacts both the exploration-exploitation balance and training stability in GFlowNets.” I feel like this statement needs a much more detailed discussion, as these challenges are not immediately obvious (past typical hyperparameter optimization).
- In my opinion, this lack of more detailed elaboration on that point is the biggest weakness of the paper. For instance, based on what I understood from the appendix, there was little attention given to beta tuning for baseline methods. As far as I understand, they all operate on the full search space, making the exact choice of beta not as critical, since the methods are compared in the exact same setting; whereas the proposed approach operates on subspaces, which would likely lead to a different choice for optimal beta. Would it be possible to demonstrate that improved performance is not due to a better choice of beta?
- The experimental component of the paper is fairly superficial. Besides perhaps Figure 4a, the remaining results essentially show the performance in terms of discovered modes across different tasks. There is little insight into why the method works (or why it doesn’t [that well], for that matter, as in e.g. RNA3 task).
- The paper has some minor formatting issues (e.g. in terms of whitespaces for citations), and some statements that frankly just don’t read much like a scientific paper (e.g. “What we found extremely strange is that (…)” - to be clear, it’s specifically about the phrasing used).

**Questions:**

- In the case of redefined base arms (Section 4.1.3), are these used exclusively for the sake of space pruning, or added as actions for the model (i.e. action space is modified)?
- What do “action values” refer to in Figure 4?
- Are the runs done across multiple random seeds? If so, why confidence intervals are not provided?
- When the term “reward” is used in e.g. Section 4.1.2, does this refer to the GFlowNet reward, or is it a terminology clash?

---

> ### Author Response · Authors · 2025-11-22
>
> Dear Reviewer 9Be3,
>
> Thanks for your recognition of our contribution to this field. Thank you very much for your valuable comments, which are crucial to the improvement of our paper. We would like to clarify your concerns point by point in the following.
>
> > W1. “Optimizing the inverse temperature parameter β presents non-trivial challenges, as its selection critically impacts both the exploration-exploitation balance and training stability in GFlowNets.” this statement needs a much more detailed discussion.
>
> Sorry for your confusion. Intuitively, the inverse temperature parameter $\beta$ controls the sharpness of the sampling distribution: when $\beta \rightarrow 0$, the policy approaches a uniform distribution (pure exploration), while large $\beta$ values concentrate probability mass around high-reward states (strong exploitation). Despite this conceptual simplicity, tuning $\beta$ effectively in practice presents several interconnected challenges.
>
> The choice of $\beta$ governs how aggressively the agent focuses on high-reward regions. A small $\beta$ facilitates exploration, ensuring sufficient coverage of the state space but leading to inefficient use of samples as many trajectories fall in low-reward regions[1]. Conversely, a large $\beta$ quickly biases the network toward known high-reward modes, which may cause premature mode collapse, the network overfits to a few peaks thus failing to discover other promising subspaces[2,4]. Finding a balance is particularly difficult in high-dimensional combinatorial spaces, where the landscape of R(x) can be extremely sparse[3].
>
> For completeness, we report in `W2` the empirical behavior of TB-GFN under different $\beta$ settings, illustrating the instability and sensitivity discussed above.
>
> [1]Malkin, N., Lahlou, S., Deleu, T., Ji, X., Hu, E., Everett, K., ... & Bengio, Y. (2022). GFlowNets and variational inference.
> [2]Nikolay Malkin, Moksh Jain, Emmanuel Bengio, Chen Sun, and Yoshua Bengio. Trajectory balance: Improved credit assignment in gflownets.
> [3]Elaine Lau, Stephen Lu, Ling Pan, Doina Precup, and Emmanuel Bengio. Qgfn: Controllable greediness with action values.
> [4]Kanika Madan, Jarrid Rector-Brooks, Maksym Korablyov, Emmanuel Bengio, Moksh Jain, Andrei Cristian Nica, Tom Bosc, Yoshua Bengio, and Nikolay Malkin. Learning gflownets from partial episodes for improved convergence and stability.
>
> > W2. Beta tuning issues. As far as I understand, baselines all operate on the full search space, making the exact choice of beta not as critical, since the methods are compared in the exact same setting; whereas the proposed approach operates on subspaces, which would likely lead to a different choice for optimal beta. Would it be possible to demonstrate that improved performance is not due to a better choice of beta?
>
> Thanks for the observation. We agree that the temperature parameter $\beta$ affects the sharpness of the learned distribution, but our performance gains are not attributable to a more favorable choice of $\beta$. In all tasks, we follow the baseline settings whenever possible: for molecule generation we use the default $\beta=8$ from the official GFlowNet implementation; for bit-sequence tasks all methods use $\beta=2$; and for RNA tasks baselines use $\beta=10$, while our method achieves slightly better performance with $\beta=20$. Importantly, CBFlowNet remains highly robust across a wide range of $\beta$ values.
>
> In our origin manuscript, we have tested our framework under different $\beta$ values. To further demonstrate that our framework is more robust for different $\beta$, we compare our framework with TB-GFN under different $\beta$.
>
> | Task     | Method    | β=1 | β=5 | β=10 | β=20 | β=50 | β=100 |
> | -------- | --------- | --- | --- | ---- | ---- | ---- | ----- |
> | **RNA1** | CBFlowNet | 110 | 125 | 134  | 135  | 115  | 115   |
> |          | TB-GFN    | 1   | 3   | 18   | 19   | 0    | 1     |
> | **RNA2** | CBFlowNet | 104 | 101 | 104  | 110  | 89   | 90    |
> |          | TB-GFN    | 1   | 1   | 21   | 18   | 0    | 0     |
> | **RNA3** | CBFlowNet | 33  | 35  | 43   | 36   | 22   | 34    |
> |          | TB-GFN    | 1   | 2   | 9    | 10   | 0    | 0     |
>
> As shown in the table above, CBFlowNet consistently discovers far more modes than TB-GFN under every tested $\beta \in {1,5,10,20,50,100}$, including regimes where its performance is less favorable. In contrast, TB-GFN fails to recover meaningful modes across all settings and even collapses when $\beta$ becomes large (e.g., 50 or 100). If the improvement were primarily due to tuning $\beta$, we would expect TB-GFN to match or surpass CBFlowNet for certain values, which never occurs. This behavior aligns with our explanation in W1: CBFlowNet’s subspace-switching mechanism prevents over-concentration on early peaks, enabling effective exploration even under different temperature scales. Thus, the performance gain arises from the proposed method itself rather than from $\beta$ selection.

---

> ### Author Response · Authors · 2025-11-22
>
> > W3. The experimental component of the paper is fairly superficial. Besides perhaps Figure 4a, the remaining results essentially show the performance in terms of discovered modes across different tasks. There is little insight into why the method works (or why it doesn’t [that well], for that matter, as in e.g. RNA3 task).
>
> In GFlowNet tasks, mode discovery and average reward are the standard metrics used to assess the quality of the learned flow. Beyond these, we also report complementary measures such as ELBO (Appendix L) and top-K Tanimoto similarity analysis (Appendix J), which evaluate how well the learned policy fits the target distribution and how diverse the generated samples are.
>
> To provide insight into why our method works, we include a detailed walkthrough in Appendix I using the bit-sequence task as an illustrative example. For other tasks, however, identifying or visualizing “optimal actions” is generally infeasible—especially in domains such as molecular design, where action quality is not interpretable. Thus, task-specific behavioral explanations are not realistically achievable, and we instead focus on explaining the mechanisms that make our method effective.
>
> Regarding the lower performance on RNA3, we already noted this in lines 430–431. RNA3 uses a reward function substantially different from RNA1/2, making mode discovery much more challenging. All baselines exhibit a significant drop on this task; nevertheless, our method still obtains the highest number of modes and average reward, albeit with slightly higher instability.
>
> For the Top-1000 reward metric, our method performs comparably to QGFN, while LSGFN achieves the best scores due to its local-search strategy, which favors finding many high-reward but structurally similar solutions. Thus, this metric reflects LSGFN’s inductive bias rather than a weakness of our method.
>
> We also clarify the instability observed. CUCB selects subspaces by balancing exploitation and exploration. Occasionally, underexplored subspaces receive high UCB values (Algorithm 1, line 11), leading CUCB to explore them even if they initially appear less promising. This behavior is necessary for discovering high-reward subspaces that would otherwise remain unvisited.
>
> Finally, we summarize the key reasons why our method works:
>
> - Efficient sampling: It filters out low-quality actions while preserving high-quality ones, improving both sampling efficiency and sample quality.
>
> - High diversity: Frequent switching between subspaces encourages exploration of dissimilar regions, yielding more diverse solutions.
>
> - Balanced exploration–exploitation: Unlike methods such as QGFN which focus solely on exploitation and risk losing important information, our approach maintains a strong balance between the two.
>
>
> > W4. The paper has some minor formatting issues (e.g. in terms of whitespaces for citations), and some statements that frankly just don’t read much like a scientific paper (e.g. “What we found extremely strange is that (…)” - to be clear, it’s specifically about the phrasing used).
>
> Thanks for pointing out the issue. We have changed the statement `What we found extremely strange is that ...` to `We observe
> that the baseline SUBTB consistently underperforms in both mode discovery and the generation of
> high-reward candidates. This pattern is not unique to our experiments; a
> similar trend was reported in the molecule design task by Lau et al. (2024), suggesting that SUBTB
> may face intrinsic challenges in effectively exploring complex reward landscapes.`
> We have also checked our manuscript for other formatting issues.

---

> ### Author Response · Authors · 2025-11-22
>
> > Q1. In the case of redefined base arms (Section 4.1.3), are these used exclusively for the sake of space pruning, or added as actions for the model (i.e. action space is modified)?
>
> Both using redefined base arms exclusively for the sake of space pruning and changing the environment (add as actions for the model) are feasible. If we use the redefined base arms as actions for the model, the network structure and action space are modified physically. However, modifying the environment would introduce additional implementation complexity and could lead to an unfair comparison, as the environments for the baseline methods remain unchanged. Therefore, in our implementation, we employ the redefined base arms exclusively for space pruning, without altering the action space.
>
> > Q2. What do “action values” refer to in Figure 4?
>
> Thanks for pointing out the issue. Action values refer to the empirical mean rewards for each base arm, corresponding to $\hat{\mu_i}$ in `eq. 4`. We will explain the definition of action value in the revised version.
>
> > Q3. Are the runs done across multiple random seeds? If so, why confidence intervals are not provided?
>
> Thank you for pointing this out. In the original submission, we did not report results averaged over multiple random seeds. Our goal in this initial submission was to provide a clear demonstration of the method’s behavior and overall trends, thus we do not optimize through different seeds.
>
> > Q4. When the term “reward” is used in e.g. Section 4.1.2, does this refer to the GFlowNet reward, or is it a terminology clash?
>
>
> We defined two different rewards: $X_i^t$ and R(x) in `Section 4.1.2`, where R(x) denotes the GFlowNet reward (or more precisely, the environment reward), while $X_i^t$ represents the reward of base arm i at round t. Thus, the meaning of the term “reward” depends on its context. For example, in the phrase “raw reward R(x) from the environment”, it refers to the environment (GFlowNet) reward, whereas in “the reward of base arm i at round t”, it specifically denotes the base arm reward.

---

### Note · Authors · 2026-01-23

I have read and agree with the venue's withdrawal policy on behalf of myself and my co-authors.